# Kombucha Versus Vegetal Cellulose for Affordable Mucoadhesive (nano)Formulations

**DOI:** 10.3390/gels11010037

**Published:** 2025-01-04

**Authors:** Ioana Popa-Tudor, Naomi Tritean, Ștefan-Ovidiu Dima, Bogdan Trică, Marius Ghiurea, Anisoara Cimpean, Florin Oancea, Diana Constantinescu-Aruxandei

**Affiliations:** 1Polymers and Bioresources Departments, National Institute for Research and Development in Chemistry and Petrochemistry—ICECHIM, Splaiul Independentei nr. 202, Sector 6, 060021 Bucharest, Romania; ioana.popa.tudor@icechim.ro (I.P.-T.); naomi.tritean@icechim.ro (N.T.); ovidiu.dima@icechim.ro (Ș.-O.D.); bogdan.trica@icechim.ro (B.T.); marius.ghiurea@icechim.ro (M.G.); 2Faculty of Biology, University of Bucharest, Splaiul Independentei nr. 91-95, Sector 5, 050095 Bucharest, Romania; anisoara.cimpean@bio.unibuc.ro

**Keywords:** nanocellulose, brewer’s spent grains, kombucha fermentation, hydrogels, cytocompatibility, gingival fibroblasts, mucoadhesion

## Abstract

Cellulose nanofibers gained increasing interest in the production of medical devices such as mucoadhesive nanohydrogels due to their ability to retain moisture (high hydrophilicity), flexibility, superior porosity and durability, biodegradability, non-toxicity, and biocompatibility. In this work, we aimed to compare the suitability of selected bacterial and vegetal nanocellulose to form hydrogels for biomedical applications. The vegetal and bacterial cellulose nanofibers were synthesized from brewer’s spent grains (BSG) and kombucha membranes, respectively. Two hydrogels were prepared, one based on the vegetal and the other based on the bacterial cellulose nanofibers (VNC and BNC, respectively). VNC was less opaque and more fluid than BNC. The cytocompatibility and in vitro antioxidant activity of the nanocellulose-based hydrogels were investigated using human gingival fibroblasts (HGF-1, ATCC CRL-2014). The investigation of the hydrogel–mucin interaction revealed that the BNC hydrogel had an approx. 2× higher mucin binding efficiency than the VNC hydrogel at a hydrogel/mucin ratio (mg/mg) = 4. The BNC hydrogel exhibited the highest potential to increase the number of metabolically active viable cells (107.60 ± 0.98% of cytotoxicity negative control) among all culture conditions. VNC reduced the amount of reactive oxygen species (ROS) by about 23% (105.5 ± 2.2% of C−) in comparison with the positive control, whereas the ROS level was slightly higher (120.2 ± 3.9% of C−) following the BNC hydrogel treatment. Neither of the two hydrogels showed antibacterial activity when assessed by the diffusion method. The data suggest that the BNC hydrogel based on nanocellulose from kombucha fermentation could be a better candidate for cytocompatible and mucoadhesive nanoformulations than the VNC hydrogel based on nanocellulose from brewer’s spent grains. The antioxidant and antibacterial activity of BNC and both BNC and VNC, respectively, should be improved.

## 1. Introduction

The study of polymers with mucoadhesive properties began in 1947 with the attempts of Scrivener and Schantz to develop an effective formulation for the delivery of penicillin [1]. Subsequently, several delivery systems were developed, e.g., hydrogels, emulsions, suspensions, films, tablets, capsules, patches, etc. Mucoadhesion is a particular physicochemical phenomenon of bioadhesion, in which the process of interfacial adsorption–adhesion takes place on a mucilaginous surface [2]. The term derives from a class of natural glycoproteins–mucins, which are produced by the epithelial tissue, being present in the areas lined by the epithelial mucosa (oral, vaginal, nasal, intestinal, etc.) and having the defining property of increased protection and lubrication [3]. The mucoadhesion of different formulations is studied in relation to different mucosal tissues to ensure a targeted and optimal release of encapsulated active substances. Moreover, the adhesion of transdermal patches or dressings for various health conditions is of interest in the biomedical field [4]. The treatment of the oral mucosal conditions requires a long-term adherence of the therapeutic agent to the target site. The adhesion may be hindered by the continuous salivary flow or the mechanical movements of the tongue. Therefore, the formulations that act as a vehicle for the delivery of different bioactive compounds/drugs to the oral mucosa must exhibit excellent mucoadhesive properties [5,6].

Mucoadhesive properties have been discovered and exploited for several oligo- and polysaccharides, proteins, and peptides. Among the most studied polymers with mucoadhesive properties used as mucosal delivery systems are cellulose, hyaluronic acid, alginate, chitosan, carbomers, and their derivatives [7,8]. Cellulose is an abundant natural polymer composed of D-glucopyranose units linked by β-1, 4-glycosidic bonds [9]. It is well known that the lignocellulosic matrices are a valuable source of cellulose. Although vegetal cellulose is nature’s most abundant biopolymer, it is difficult to obtain nanocellulose from complex vegetal matrices where it is combined with hemicellulose, pectin, and lignin in order to form the plant cell wall. As a result, the production of cellulose by bacteria and microbial fermentative processes, e.g., using a symbiotic culture of bacteria and yeasts (SCOBY) such as the one from kombucha, gained a great interest. This is due to the specific physicochemical properties that result from the biosynthesis process and that influence the biological activities.

Vegetal and bacterial nanocellulose represent promising biopolymers for the biomedical field due to their biocompatibility, hydrophilic nature, biodegradability, non-toxicity [10], water retention capacity [11,12], superior flexibility and porosity, and increased durability [13]. Due to the macromolecular chains that present a high number of -OH and -COOH functional groups which are responsible for the ability of nanocellulose to adsorb and retain water, it can form hydrogels by establishing new inter-chain hydrogen bonds. The structure of hydrogels can be irreversible in the presence of certain cross-linking agents or reversible through physicochemical interactions [7]. Reversible hydrogels have been shown to exhibit superior mucoadhesive properties than irreversible hydrogels due to the high mobility of the chains. A suitable hydrogel for biomedical applications should exhibit a sol–gel transition under specific conditions without releasing toxic byproducts or having a negative impact on the surrounding tissues [14]. The cells can also be incorporated into the hydrogels creating 3D cellular scaffolds that lead to tissue regeneration [15,16].

Bacterial (nano)cellulose was shown to have some distinct properties compared with the vegetal (nano)cellulose, such as higher purity, higher crystallinity, mechanical stability, higher water-holding capacity, and less intensive purification processes than plant (nano)cellulose [17,18,19,20,21,22]. Plant (nano)cellulose has the advantage of antimicrobial, antioxidant, and anti-inflammatory properties compared to bacterial (nano)cellulose [18,19,20]. It is generally considered that bacterial (nano)cellulose is more suitable for biomedical applications, but there are limited studies to directly compare bacterial (nano)cellulose with vegetal (nano)cellulose for biomedical applications under the same conditions. In a study from 2013, bacterial cellulose was tested for dental canal treatment and proven to be superior to paper points (plant cellulose), which is a conventional material [23].

In the light of the above, the aim of this study was to compare the capacity of selected bacterial and vegetal nanocellulose to form mucoadhesive and cytocompatible hydrogels. Two types of cellulose nanofiber-based hydrogels, respectively, a hydrogel based on vegetal nanocellulose from brewer’s spent grains (VNC) and a hydrogel based on bacterial nanocellulose from kombucha fermentation (BNC), were compared. Brewer’s spent grains (BSG) represent a lignocellulosic by-product from the beer industry that can be exploited for the recovery of cellulose, lignin, carbohydrates, glycoproteins, and polyphenols [24]. We selected BSG as it is one of the most abundant lignocellulosic by-products which can be collected from a concentrated source [25]. The large quantities of BSG that are produced worldwide allows and moreover requires valorization at industrial scale. Kombucha is a known fermented, sweetened black tea that produces a thick bacterial cellulose membrane as a by-product, which represents a more cost-effective source of bacterial cellulose than a single bacterial strain. By recycling the residual materials, value-added biocompounds/biomaterials can be obtained for several applications, i.e., biotechnological or pharmacological products, medical devices, etc. [26]. Physicochemical properties, hydrogel–mucin interaction, and the resulting complexes (VNCMu and BNCMu, respectively) and the cytocompatible behavior were characterized in order to assess the hydrogels as potential candidates for various biomedical applications.

## 2. Results and Discussion

### 2.1. Morphological and Structural Aspects of Nanocellulose-Based Hydrogels and Hydrogel–Mucin Systems

A schematic representation of the main steps involved in the preparation of the two nanohydrogels (VNC and BNC from BSG and kombucha fermentation, respectively) is depicted in Figure 1. As can be seen, both types of cellulose have a whitish appearance which indicates high purity upon purification (Figure 1b,d). The hydrogels have a translucid and similar appearance. The BNC hydrogel has a slightly more opaque appearance than the VNC hydrogel (Figure 1e,f). Moreover, the BNC hydrogel had a more defined gel texture than the VNC hydrogel, which was more fluid. After the freeze-drying process, the VNC hydrogel is fluffier and more fragile than the BNC hydrogel which has a flexible and resistant texture (Figure 1g,h).

It was previously reported that by alkaline treatment, followed by bleaching and microfluidization, lattice-type cellulose nanofibrils were obtained from brewer’s spent grains, with at least 10 microfluidization cycles being necessary in order to produce a uniform fibrillar structure, as evidenced by SEM analysis [27]. In our case, 10 cycles induced defibrillation, but the structure was not sufficiently uniform. Moreover, in the case of the bacterial cellulose from kombucha, we previously tested 1, 10, and 20 microfluidization cycles. Defibrillation of the microfibrillar structure of the bacterial cellulose into nanofibrils was obtained after 20 microfluidization cycles based on the XRD analysis [28]. Therefore, in the present study, we used 20 cycles of microfluidization in order to obtain the bacterial as well as the vegetal nanocellulose. TEM analysis highlighted the fibrillar structure of the two types of nanocellulose-based hydrogels. Nanomaterials are defined as materials with at least one of their dimensions in the nanometric scale, i.e., ≤100 nm, sometimes higher; a definition which was adopted for cellulose as well [29,30,31,32,33,34,35]. These are technological definitions that are different from the biological term microfibril with nm diameters [35]. As can be seen in Figure 2, the bacterial cellulose was apparently more efficiently nano fibrillated than the vegetal cellulose after 20 cycles. The reason could be that the bacterial cellulose is naturally produced as nanoscale fibrils, whereas the extracted plant cellulose was reported to have 13–22 µm [30]. Nevertheless, one should consider as well that the fibrils have the tendency to aggregate due to the -OH groups and concentrating the solutions to form hydrogels accelerates this process [36]. The aggregation could be favored by the drying process during TEM acquisition as well [30]; therefore, we shall continue to use the term nanofibers for the vegetal nanocellulose in the following. The nanofibers of the BNC hydrogel interweave into a dense network-like structure (Figure 2d,e,f). The type of biomass can lead to significant differences in the structure of nanocellulose. Cellulose nanofibers from poplar wood exhibited larger diameters than those from rice straw and a looser arrangement, as revealed by TEM analysis [37]. In our previously reported study, by bleaching of bacterial cellulose with sodium hydroxide followed by colloidal mill grinding and 25 cycles of microfluidization, TEM analysis revealed the formation of a uniform mesh-like structure, with the diameter of the cellulose nanofibrils of approx. 10–50 nm. By using the spray-drying process instead of microfluidization, fibrillar bundle or ribbon-like structures of nanocellulose were obtained [29], suggesting the development of different cellulosic nanostructures based on the mechanical destructuration technique.

The SEM analyses of VNC and BNC hydrogels highlighted some differences in the structure and the arrangement of the two types of cellulose nanofibers. The secondary electrons (SE) detector was used in order to investigate the topography of the samples (Figure 3), and the backscattered electrons (BSE) detector provided structural insights from a larger depth (Figure 4). The VNC hydrogel exhibited a more relaxed arrangement of cellulose fibers (Figure 3a and Figure 4a) in comparison with the mesh-like structure of BNC (Figure 3b and Figure 4b).

Following the hydrogel–mucin interaction, a relatively compact structure was obtained for both VNCMu (Figure 3c and Figure 4c) and BNCMu (Figure 3d and Figure 4d), which is similar to the morphological aspect of mucin (Figure 3e and Figure 4e), but with some protuberances. The mucoadhesive properties of vegetal cellulose nanofibers/nanocrystals and the morphological characteristics of the complexes have been previously analyzed not by SEM, but by confocal microscopy [38]. The authors mixed a suspension of Fluorescent Brightener 28-labeled nanocellulose/nanocrystals from bleached softwood kraft pulp with acridine orange-labeled pig mucus, followed by rinsing of the hydrogel–mucin system with digestive juice. A significant adhesion of both nanocellulose and nanocrystals to the porcine mucosal layer in simulated gastric and intestinal fluids was observed. Moreover, due to the fibrillar structure of the nanocellulose, cellulose nanofibrils were entangled within the mucus layer, whereas in the case of the cellulose nanocrystals, a more homogeneous structure was obtained, the nanocrystals being spread over the mucus layer.

The wide-angle X-ray scattering (XRD-WAXS) results of vegetal nanocellulose and bacterial nanocellulose before and after its interaction with mucin are presented in Figure 5. The diffraction spectra were compared with the diffraction patterns of known cellulose allomorphs, available in the PDF5+ PDXL Rigaku database, respectively, one-chain triclinic Iα (PDF card No. 00-056-1719), two-chain monoclinic Iβ (PDF card No. 00-060-1502), and amorphous cellulose (PDF card No. 00-060-1501). One-chain Iα allomorph prevails in bacterial cellulose and green algae, the reason for which it is known as bacterial-algal cellulose, or *Acetobacter-Valonia* cellulose [39,40,41]. Two-chain Iβ allomorph prevails in higher plants and it is known as cotton-ramie cellulose. The chemical and crystal structures of Iα and Iβ allomorphs have been previously predicted and shown in numerous studies [42,43,44]. VNC from BSG showed a main diffraction peak at approx. 22.60°, corresponding to cellulose Iβ, and a predominantly amorphous character attributed to amorphous cellulose, with the crystallinity index (Xc) determined to be 35% (Figure 5a).

The diffractogram of BNC presented in Figure 5b evidenced the crystallographic patterns of cellulose Iα with the characteristic peak at 16.76°. The first peak at 14.50° represents the convolution of peak Iα at 14.26° and peak Iβ at 14.83°. Similarly, the main peak at 22.66° is a convolution between peak Iα at 21.8° and peak Iβ at 22.71°. The biosynthesis conditions of present BNC were similar with the ones previously reported [29,45], therefore, the spectra are similar in peak position and relative intensities. Under different fermentation conditions, the relative peak intensities, correlated with the Iα:Iβ ratio, suffer particular changes [46]. It has been shown that static conditions of cultures favor the content of cellulose Iα, whereas under agitated conditions, cellulose Iβ seems to be favored [43].

The WAXS diffractogram of mucin showed two main peaks at approx. 20° and 21.42° and a highly amorphous character, with 14% Xc. Individual diffractograms of mucin are apparently scarce since mucin is generally considered completely amorphous [47,48], although in some cases it showed a sharp peak around 27° [49]. The mucin spectrum in Figure 5 is somewhat similar with the one considered completely amorphous, with the main peak at approx. 22° and additional small peaks [47]. In one of our previous studies [50], the same mucin showed a similar diffractogram with the one in Figure 4, but with the two main peaks at approx. 19.50° and 21.52° and two additional small peaks at approx. 6° and 9°, which could be related to the crystallization/lyophilization step. In another study, by small-angle X-ray scattering (XRD-SAXS) it was evidenced that MUC5AC mucin at 1 mg/mL concentration and pH 7 had a gyration radius of 42 ± 4 nm, whereas at the boundary gelation pH of 3.5, the gyration radius decreases to 23 ± 3 nm [32].

The physical interaction between VNC and mucin led to the predominantly amorphous system VNCMu with 16% Xc and with four small diffraction peaks around 2θ angles 15°, 18.18°, 19.62°, and 21.52°. The new peaks suggest particular macromolecular interactions with the amorphous cellulose fragments and Iβ chains, which leads to a new distinct structure. In Figure 5b, the crystallinity of BNCMu (33%) is higher than that of VNCMu which indicates a more ordered structure of BNCMu than of VNCMu. The shifted peaks at 17.22° and 21.54° reveal the mucin interaction with the cellulose Iα chains, and the shifted peak at 19.44° suggests the interaction with the amorphous cellulose. The IαIβ composite structure at 22.66° appears partially covered, but unshifted.

FTIR recordings of nanocellulose spectra reveal distinct molecular vibrations in the structures of the two types of nanocellulose. Figure 6 presents the similarities and differences between the two structures (VNC and BNC). The broad band at 3343 cm^−1^ (VNC, BNC), corresponding to the stretching vibrations of the OH bonds, attests to the inter- and intramolecular hydrogen bonds found in the beta-glycosidic cellulose chains [51]. This band is apparently a little more intense in the VNC structure. The two bands at 2912 cm^−1^ (VNC) and 2897 cm^−1^ (BNC), which describe the symmetric/asymmetric stretching vibrations of the C-H bonds attached to the glucose rings in the cellulose structure, are distinct and more intense in VNC. The bands at 1633 cm^−1^ (VNC) and 1622 cm^−1^ (BNC), which can describe the stretching vibrations of water molecules physiosorbed on the hydrophilic surface of nanocellulose, are broad but distinct. The band specific to BNC is more intense, likely due to its slightly higher hydrophilic character compared to VNC. The band at 1427 cm^−1^, corresponding to the shear/bending vibrations of the -CH_2_ bonds, is found in both nanocelluloses, but presents some peculiarities: in VNC, it is broader, whereas in BNC, it is sharper and more intense. This behavior could be related to the lower crystallinity of VNC compared to BNC. The bands at 1367 and 1317 cm^−1^ (VNC)/1364 and 1315 cm^−1^ (BNC), which correspond to the in-plane deformation vibrations of the -CH bond in the glucose units, are shifted due to the different bending modes of the two structures, which exhibit different in-plane arrangements. The bands at 1200 cm^−1^ (VNC)/1202 cm^−1^ (BNC), which are associated with the stretching vibrations of the C-C and C-O bonds within the glycosidic ring, present both similarities and differences. They are small and of low intensity but are slightly shifted, probably influenced by intra- and intermolecular interactions. The region from 1200 to 1000 cm^−1^ is characteristic of glycosidic bonds. The shoulders at 1157 and 1105 cm^−1^ (VNC, BNC) describe the stretching vibrations of ether bonds (C-O-C, connecting the β(1→4) glycosidic units) and C-O bonds of primary alcohols (possibly located at the C6 position of glucose) [52]. The band at 1055 cm^−1^ (VNC) suggests stretching vibrations of the C-OH bond in the β-glucose structure or groups attached to the glycosidic ring. In the BNC spectrum, two convoluted bands at 1051 and 1024 cm^−1^ indicate stretching vibrations of the C-OH bonds in the primary and secondary alcohols within the BNC structure [53]. In the fingerprint region, the band at 899 cm^−1^ is preserved, indicating the presence of the glycosidic structure. In the low wavenumber region, there are three main bands of which two bands are shifted: 665, 607, and 561 cm^−1^ (VNC)/665, 611, and 559 cm^−1^ (BNC), likely due to the different vibrations of the out-of-plane bonds.

The Mu spectrum is complex, reflecting both hydrophilic (carbohydrates) [54] and hydrophobic (peptides, lipids) regions. The broad band at 3258 cm^−1^ corresponds to the stretching vibrations of the -OH groups present in the carbohydrates and NH group of amide bonds in Mu. The broadness indicates strong hydrogen bonds between these groups. The band at 3061 cm^−1^ describes the stretching vibrations of the C-H bond in the amino acids that make up the proteins in Mu. The bands at 2924 and 2855 cm^−1^ attest to the stretching vibrations of the C-H bonds in the aliphatic -CH_2_ and -CH_3_ groups. The vibrations of the bands at 1632 (amide I) and 1521 cm^−1^ (amide II) are associated with the protein content. They reflect the stretching vibrations of the peptide bond C=O [55] and the N-H in plane bending associated with the C-N stretching, respectively. The bands at 1452 and 1400 cm^−1^ could come from the deformation vibrations of the -CH_2_ and -CH_3_ groups and stretching vibrations of the C-O bond in the carboxylic group, respectively. The band at 1340 cm^−1^ corresponds to the deformation vibrations of the C-H/O-H bond in the carbohydrates in Mu. The band at 1227 cm^−1^ could come from the stretching vibrations of the C-O bond in the carbohydrates in Mu. The region between 1450 and 1300 cm^−1^ is also known to have contributions from the amide III region in proteins. The band at 1036 cm^−1^ is characteristic of carbohydrates in Mu, describing the stretching vibrations of the C-O bond. The small band at 810 cm^−1^ may suggest deformation vibrations of the C-H bonds in the aromatic ring or out-of-plane vibrations in carbohydrates. The band at 523 cm^−1^ may be attributed to the skeletal vibrations of glycosidic bonds in carbohydrates or to inorganic components, such as sulfur.

Mucin interacts slightly differently with the two nanocelluloses due to differences in crystallinity and hydrophilicity, as highlighted by the IR spectra through shifted bands, modified band appearances, or varying intensities. As shown in Figure 6, the bands in the Mu spectrum predominate.

As expected, mucin in association with VNC and BNC exhibits specific hydrophilic interaction bands. In the high wavenumber region, some band changes are observed (Figure 6a). The band at 3258 cm^−1^ (Mu) shifts to 3260 cm^−1^ (VNCMu) and becomes more intense. The bands at 3061 and 2924 cm^−1^ (Mu) shift to 3076 and 2920 cm^−1^ (VNCMu) and decrease in intensity. The band at 2855 cm^−1^ is preserved but decreases in intensity. In the mid-wavenumber region, the bands at 1632, 1521, and 1452 cm^−1^ (Mu) shift to 1634, 1537, and 1450 cm^−1^ (VNCMu), respectively. The band at 1400 cm^−1^ is preserved but decreases in intensity. The modification of the band at 1340 cm^−1^ (Mu) is more significant, with an increase in intensity and a shift to 1319 cm^−1^. The band at 1227 cm^−1^ is preserved but shifts to 1229 cm^−1^. The band at 1036 cm^−1^, specific to carbohydrates, is preserved and shifts to 1041 cm^−1^. The small band at 810 cm^−1^ flattens, and a new small band appears at 878 cm^−1^. The band at 523 cm^−1^ is preserved. These spectrum modifications demonstrate that the association between Mu and VNC mainly involves hydrophilic groups, enabling the mucoadhesive interaction between Mu and the nanocellulose hydrogels.

The same behavior is observed for Mu in association with BNC, where the hydrophilic groups of Mu interact with the hydrophilic groups of BNC (Figure 6b). Due to the different structure of BNC compared to VNC, the bands shift to different wavelengths but maintain a similar affinity for the hydrophilic regions as in VNC. It is worth mentioning the opposite band shifts of the amide I band (1632 cm^−1^) and the main carbohydrate band (1036 cm^−1^) in Mu to 1634 cm^−1^ and, respectively, 1041 cm^−1^ in VNCMu with respect to 1630 cm^−1^ and, respectively, 1034 cm^−1^ in BNCMu.

### 2.2. Physicochemical Properties of Nanocelullose-Based Hydrogels and Hydrogel–Mucin Systems

The VNC hydrogel and BNC hydrogel presented a surface tension of 50.44 ± 0.66 mN/m and 55.29 ± 0.29 mN/m, respectively. A higher surface tension, as shown for the BNC suspension, correlates with a higher stability of the suspension and also with a higher resistance to flow deformation [56]. A lower surface tension, as for VNC suspension, correlates with an easier separation of the hydrophilic/hydrophobic cellulose fragments. The cellulose concentration, electrostatic charge, and fiber length influence the surface tension. A 5% microcrystalline cellulose suspension at pH 4.0, with rod dimensions of 180 ± 30 nm length and 8 ± 3 nm diameter, showed a surface tension of ca. 66.2 ± 0.5 mN/m [56]. VNCMu recorded a significant decrease of 8.37 ± 0.78% in the surface tension compared with VNC. The decrease (12.28 ± 0.97%) in surface tension of BNCMu with respect to BNC was higher (σ = 0.006) in comparison with the decrease in VNCMu. Mu showed the lowest surface tension among the samples, i.e., 39.33 ± 0.47 mN/m and induced the decreases mentioned above.

The VNC hydrogel had a contact angle of 52.70 ± 0.56° and 56.90 ± 0.40° on the hydrophilic surface (polar, glass) and on the hydrophobic (non-polar, polystyrene) surface. This suggests an amphiphilic nature of VNC, with a small tendency towards hydrophilicity. The amorphous anisotropic cellulose is known to be more hydrophobic [56]. In the case of VNCMu, the contact angle on the hydrophilic surface decreased by 3.42 ± 0.68% and on the hydrophobic surface increased by 2.18 ± 0.73% compared with VNC. The BNC hydrogel presented a contact angle of 48.27 ± 0.57° and 65.13 ± 0.35° on the hydrophilic surface and on the hydrophobic surface, respectively. This indicates a more hydrophilic character of BNC compared with VNC. For BNCMu, a slightly higher decrease in the contact angle on the polar surface (3.59 ± 1.14%) as well as a significantly higher increase (σ = 0.01) on the non-polar surface (4.40 ± 0.41%) were observed compared to the decrease, respectively, and increase, recorded in the case of VNCMu. The contact angle on both surfaces was determined for the mucin suspension as well. Due to the predominantly hydrophilic nature of mucin [33], the contact angle on the hydrophilic surface was 45.67 ± 0.71° and on the hydrophobic surface the contact angle was significantly higher, i.e., 78.27 ± 0.45° (Table 1).

BNC had a significantly higher mucin binding efficiency (92.85 ± 2.62% and 82.18 ± 2.22% at a nanocellulose/mucin ratio (*w*/*w*) of 12 and 4, respectively) compared to VNC (74.45 ± 2.54% and 46.66 ± 1.47% at a nanocellulose/mucin ratio (*w*/*w*) of 12 and 4, respectively), as shown in Figure 7. This may be due to the higher hydrophilicity of bacterial nanocellulose, which suggests a higher availability of hydroxyl groups. This is indicated by the FTIR analysis as well, which shows that the -OH and -NH groups are more involved in inter-molecular interactions and therefore less available for hydrogen interactions with mucin. There is a stronger interaction of the bacterial nanocellulose and the mucin glycoproteic structure induced by more free -OH groups than in VNC. The higher availability of the free -OH groups correlates with Iα cellulose allomorph found in BNC and with the cellulose fragments that compose the amorphous cellulose. The stronger Iβ allomorph, predominant in VNC, has a higher number of hydroxyl groups involved in hydrogen bridges between the two chains [29]. Additionally, the fibrillar structure of BNC is more developed, with longer chains compared to VNC [29,57]. The longer nanofibrils entangle with the branched structure of mucin and therefore increase the mucoadhesion.

The viscoelastic behavior of the VNC and BNC suspensions observed by the rheological experiments performed in frequency sweep, flow sweep, and axial mode are depicted in Figure 8. The oscillatory or frequency sweep mode presented in Figure 8a,b allows the evaluation of storage modulus G’, loss modulus G”, complex viscosity η*, dynamic viscosity η’, and phase angle δ in the chosen range of 0.1–100 rad/s angular frequencies ω. The storage modulus describes the elastic contribution, meaning the solid-like behavior of the suspension, whereas the loss modulus describes the viscous contribution, meaning the liquid-like behavior. The complex modulus |G*| describes the overall resistance to deformation of the suspensions, including the elastic, recoverable deformation and the viscous, non-recoverable deformation, and if divided by the angular frequency ω, it gives the complex viscosity η*. Both VNC and BNC suspensions show in Figure 7a,b a higher storage modulus than loss modulus, this fact being generally considered a gel property [58]. BNC showed higher storage and loss modulus values than VNC, respectively, a mean G’ = 28.26 Pa and G” = 3.92 Pa at 4 rad/s for BNC compared with a mean G’ = 0.88 Pa and G” = 0.13 Pa at 4 rad/s for VNC. In another study, a 1% microcrystalline-derived nanocellulose suspension of 211 ± 114 nm fibrils showed a storage modulus of approx. 40 Pa and a loss modulus of approx. 9 Pa [59]. Another difference between VNC and BNC consists in the storage modulus exponential, which increased for VNC at angular frequencies higher than 10 rad/s, up to 18.67 Pa at 100 rad/s, whereas the BNC storage modulus slowly increases up to 59.83 Pa at ω = 100 rad/s. The loss modulus G” showed a sinusoidal increasing trend for both VNC and BNC, with a maximum value of G” = 0.26 Pa at 25.12 rad/s for VNC, respectively, and G” = 5.75 Pa at 63.10 rad/s for BNC. The initial complex viscosity of BNC (η* = 213.66 Pa∙s) is approx. 33 times higher than that of VNC (η* = 6.48 Pa∙s). The complex viscosity of BNC suspension constantly decreases with the increase in the angular frequency ω. This behavior evidences a shear thinning (pseudoplastic) behavior, which typically suggests the orientation and disentanglement of the nanofibers [59]. The VNC suspension is pseudoplastic up to 25.12 rad/s and becomes shear thickening (dilatant) at higher ω, probably because of amorphous cellulose disordering and fragment intercalation. The oscillatory hysteresis loop is not significant, showing only a slight thixotropic behavior for VNC. This is evidenced as a small decrease in the complex viscosity on the reverse ramp, a small decrease in the storage modulus, and a small increase in the loss modulus and in phase angle δ. High values of phase angle δ correlate with a viscous, liquid behavior, whereas low values of δ suggest an elastic, solid-like behavior [50].

At higher shear rates in flow sweep mode depicted in Figure 8c,d, both nanocellulose suspensions show a complex behavior with strong hysteresis. The steady state viscosity η decreases in the shear rate γ* range of 1–4 1/s for VNC from 3.03 Pa∙s to 0.08 Pa∙s, after which increases to a maximum of 0.27 Pa∙s at γ* = 10 1/s and decreases again to 0.01 Pa∙s at γ* = 100 1/s. A critical shear rate around γ* = 5 1/s was previously observed for other nanocellulose suspensions [59]. For BNC, η constantly decreases from 37.10 Pa∙s at γ* = 0.1 1/s to 0.01 Pa∙s at γ* = 100 1/s. The viscosity of vegetal nanocellulose suspensions is known to be very high due to fibril length and entanglement. A 0.7% suspension of fibrils with a medium length of 820 ± 570 nm showed a viscosity of approx. 4 Pa∙s, whereas a 1% suspension of 211 ± 114 nm fibrils showed a viscosity of approx. 2 mPa∙s at γ* = 1 1/s [59]. The hysteresis loop is significant for both nanosuspensions, showing a stronger thixotropy for VNC than BNC. The thixotropy of nanocellulose water suspensions depends on the structural hierarchy and concentration of the nanoparticles, some possible hierarchies being individual nanoparticles, mesophase liquid crystalline domains, chiral nematic, and nematic structures [58].

The thixotropic behavior is more pronounced at higher concentrations [58] and for the macromolecular structures with higher molar weights, whereas the rheopectic or anti-thixotropic behavior generally appears in amphiphilic systems due to repeated disequilibrium states between hydrophilic and hydrophobic components induced by shearing. The thixotropy is proportional with the hysteresis loop [58], VNC suspension being more thixotropic than BNC, as can be seen in Figure 8c,d. The Carreau–Yasuda rheological model fits best the viscosity variation for both VNC and BNC suspensions, whereas the Herschel–Bulkley fitting model describes a Bingham fluid with yield stress above which the hydrogel starts to flow, the yield stress being eight times higher for BNC (0.50 Pa) than for VNC (0.06 Pa). The yield stress can be considered as a stability parameter and as an intrinsic resistance to flow deformation, and it can be correlated with the surface tension, higher for BNC than for VNC, as previously determined.

The Carreau–Yasuda model is a continuous model for all shear rates that describes a non-Newtonian fluid with viscosity plateau at low (η_0_) and high (η_∞_) shear rates and with a power law variation in between [60,61]. The Carreau rheological model was able to fit simultaneously the simple shear, complex viscosity, stress growth, and stress relaxation functions [62], while the extension of Yasuda with a power law index provided a better transition from the Newtonian plateau to the power law region [63].

The adhesion force determined in Figure 8e,f as the axial force that opposes the detachment with 10 µm/s lifting speed of the cylindrical geometry from the nanosuspensions shows a higher value of 0.154 N for BNC compared with 0.128 N for VNC. The adhesion time is higher for VNC, with 46 s adhesion time at maximum axial force and 524 s total adhesion time, in comparison with 29 s adhesion time at maximum axial force and 467 s total adhesion time for BNC, which suggests more contact points with the quartz and geometry surfaces for VNC than for BNC. The speed of detachment evaluated as the exponential parameter c is higher for VNC than for BNC, the axial experiment conclusively suggesting multiple short nanofibrils in VNC in comparison with fewer but longer nanofibrils in BNC, considering the same cellulose concentration.

These results correlate with the XRD, surface tension, and contact angle results. Less fibrillated and more inhomogeneous VNCs together with shorter and multiple VNC fibrils correlate with an amorphous character, lower surface tension, amphiphilic character due to exposed hydrophobic glucose rings, and methyl-rich chain ends [64,65] and multiple surface adhesion points, both hydrophilic -OH functional groups and hydrophobic glucose rings. Longer and fewer BNC nanofibrils correlate with a higher crystallinity, higher surface tension due to chain length, higher hydrophilicity, more cohesive points through glucose ring–ring Van der Waals interactions combined with intra- and intermolecular hydrogen bonds, and, respectively, fewer surface adhesion points inducing a lower adhesion time.

The rheology of mucin mixtures with VNC and BNC nanosuspensions is depicted in Figure 9. Mucin is a high-molecular-weight branched glycoprotein and is rich in different functional groups, being able to establish many hydrogen bonds with all types of compounds and surfaces. The mucin rheology depends on source, pH, concentration, temperature, time, and shear rate [50,66,67]. At a pH lower than 4, mucin has a gel or solid-like behavior with G’ > G”, and at higher or neutral pH, mucin has a viscous liquid-like behavior [50,67]. In the oscillatory shearing mode presented in Figure 9a,b, VNCMu showed a viscoelastic behavior with G’ close to G” up to ω = 2.51 rad/s, with an angular frequency from which the elastic behavior becomes dominant. BNCMu is more elastic than VNCMu, with G’ higher than G”, due to the BNC influence having longer cellulose nanofibrils.

In flow sweep experiments presented in Figure 9c, the VNCMu suspension shows a rheopectic behavior, with viscosity increasing on the reverse curve. In Figure 9d, the BNCMu suspension shows a complex thixotropic behavior, with a critical shear rate around 4 1/s. Both suspensions with mucin present a yield stress determined by the Herschel–Bulkley model and it is higher for BNCMu than for VNCMu.

The axial experiments presented in Figure 9e,f evidence the adhesion force of VNCMu (Fad = 0.197 N) to be higher than that of BNCMu and the initial VNC suspension, which suggests a stronger interaction between BNC and Mu than between VNC and Mu. The adhesion times decrease in the cellulose–mucin system, suggesting that a part of the initial adhesion energy of VNC and BNC suspensions induced by the multitude of hydroxyl groups was transformed in cohesion energy with the glycoproteic functional groups of mucin.

The porosity analysis by nitrogen adsorption, presented in Figure 10, evidenced a type II adsorption isotherm for the adsorption branch of a freeze-dried BNC sample [68,69,70], correlated with non-porous or macroporous nanomaterials. Probably due to a more pronounced non-porous or macroporous character, the VNC sample did not show relevant BET results. It is probably related as well to the lower fibrillated degree of VNC compared to BNC. This could be one of the explanations for the higher affinity of BNC versus VNC towards mucin. Natural fibers are known to present low surface area, found to be around 0.5 m^2^/g for cellulose, flax, and hemp fibers [70]. Freeze-dried BNC showed a 44.7 m^2^/g specific surface area by BET analysis presented in Figure 10a, respectively, and 39.1 m^2^/g by QS-DFT analysis presented in Figure 10b. The total pore volume at standard temperature and pressure (TPV@STP) calculated on the adsorption branch for the gaseous phase was 28.5 cm^3^/g. The cumulative pore volume (CPV) calculated on the desorption branch for the condensed nitrogen was 0.044 cm^3^/g by the BET method, respectively, and 0.037 cm^3^/g by the QS-DFT method, complying with the range of 20–50 mm^3^/g characteristic for natural fibers [70]. The pore distribution is better described by DFT in Figure 10b and presents a micropore population with the mean pore diameter of 1.3 nm accounting for 37.8% specific pore volume, followed by two mesopore populations around 13.1 nm and 19.2 nm pore diameter summing and 54.1% specific pore volume. Macropores with D > 40 nm account for 8.1% from the total pore volume as determined by DFT, or around 20% as determined by BET. Small surface area and pore volume are correlated with long cellulose fibers that are mainly non-porous and/or present large macropores, the morphological properties being additionally correlated with the purification conditions [71].

The VNC hydrogel exhibited statistically significant antioxidant activity compared to the BNC hydrogel, which showed no antioxidant activity, as can be seen in Table 2 (σ = 0.01 by DPPH method and σ < 0.001 by PFRAP method). One explanation for the AOA of VNC could come from possible residual impurities that remained attached to the extracted vegetal cellulose, such as polyphenols and lignin, known for its antioxidant activity. It has been previously shown that more impure plant cellulosic/polysaccharide extracts have higher AOA activity [72]. Considering the complex and strongly bound lignocellulosic structure of BSG, it is probably more difficult to purify the cellulose from BSG than the cellulose from kombucha membrane, which is more loosely impregnated with polyphenols and melanoidins, from black tea. Another possible explanation is that the vegetal cellulose is characterized by shorter disordered cellulose chains, and consequently, by more cellulose-reducing ends [73].

### 2.3. The Cytocompatible Behavior of Nanocellulose-Based Hydrogels

A high degree of cytocompatibility of BNC was observed at all tested concentrations in comparison with the negative cytotoxicity control (C−, untreated cells). Significant increases in the number of metabolically active viable cells were recorded at the lowest concentrations of BNC tested, i.e., 104.48 ± 0.31% of C− at 0.0125% (*w*/*v*) BNC and 107.60 ± 0.98% of C− at 0.025% (*w*/*v*) BNC (Figure 8a). At concentrations between 0.0125 and 0.05% (*w*/*v*), VNC exhibited a high degree of cytocompatibility, with a significant increase in the cell viability at the lowest concentration tested (104.67 ± 1.02% of C− at 0.0125% VNC). Higher concentrations of VNC led to a statistically significant reduction in the cell viability, i.e., 94.38 ± 0.61% of C− at 0.1% (*w*/*v*) VNC and 89.36 ± 0.74% of C− at 0.2% VNC (Figure 11a).

Thus, at the lowest concentration tested, i.e., 0.0125% (*w*/*v*), there are no significant differences between the two hydrogels, both being equally efficient with respect to the cytocompatible behavior. By increasing the concentration, only the hydrogel of bacterial nanocellulose led to a further significant increase in the cell viability, showing its high potential to support tissue regeneration. The results from Figure 11a obtained after performing the CCK-8 assay can be correlated with the fluorescence microscopy images shown in Figure 11b–m, acquired after performing the LIVE/DEAD assay.

Fluorescence microscopy images acquired after labelling of the actin filaments and the nuclei show no changes in cell morphology following treatment with 0.025% (*w*/*v*) VNC and BNC (Figure 12b,c), compared with the negative cytotoxicity control (Figure 12a). The cytoskeleton is well organized in a fibrillar structure with numerous actin filaments, with the cells maintaining their characteristic phenotype.

By morphological analyses, we highlighted that the BNC hydrogel exhibited a mesh-like structure more fibrillated into relatively long fibers than VNC. It was previously demonstrated that the length of the nanocellulose fibers represents an important factor with respect to the cytocompatible behavior [74]. In the Hepa 1–6 and KUP5 liver cells, the long nanocellulose fibers with the chain length of about 6000–7000 nm presented a high degree of cytocompatibility at concentrations between 25 and 200 µg/mL. There was only a slight decrease in the viability of the KUP5 cells at the highest concentration tested. Short nanocellulose fibers with the chain length of about 100–600 nm induced significant decreases in the viability of KUP5 cells, especially at the highest concentrations tested by the authors. Hepa cells 1–6 were affected only by the short fibers of about 175 nm, pointing also to a cytotoxic effect which is correlated with the cell type [74]. Moreover, long nanocellulose fibers organized in a mesh-like structure can support the adhesion, growth, and differentiation of the cells, leading to an increase in the number of metabolically active viable cells, promoting tissue regeneration [75]. The higher porosity and higher hydrophilicity of BNC induced by more free -OH compared with VNC contribute additionally to the higher cytocompatibility of BNC compared to VNC.

A decrease in the ROS level of approx. 12% was observed at 0.025% (dry weight) BNC (120.2 ± 3.9% of C−) compared to C+ (135.5 ± 3.0% of C−). At the same concentration, the VNC hydrogel reduced the ROS level by approx. 23% (105.5 ± 2.2% of C−) compared to C+, the amount of ROS reaching the C− level and re-establishing the ROS homeostasis (Figure 13a). The quantitative analysis from Figure 13a can be correlated with the fluorescence microscopy images from Figure 13b–e acquired after labeling of the total intracellular ROS with H_2_DCFDA.

As mentioned for the chemical AOA (DPPH and PFRAP), one contribution to the higher in vitro AOA of VNC compared to BNC could come from residual antioxidant impurities in VNC, such as polyphenols that are very probable to have remained in the extracted vegetal cellulose. The other explanation mentioned for DPPH and PFRAP, of more reducing ends in VNC than in BNC, due to shorter chains, could be available in this case as well. In a previously reported study, higher crystallinity of nanocellulose was correlated with a higher induction of ROS [76]. In our case, VNC had lower crystallinity than BNC; therefore, its own contribution to ROS generation would be also lower. This could represent an additional factor leading to a lower ROS level by the VNC treatment than by the BNC treatment. Nevertheless, BNC presented some in vitro AOA as well, which could come from some reducing residual polyphenols or condensed polyphenols (melanoidins) resulting from the black tea of the kombucha culture. Melanoidins were previously shown to have antioxidant properties [77,78,79].

The VNC and BNC hydrogels did not show antibacterial activity against the Gram-positive (*B. cereus*, *E. faecalis*, and *S. aureus*) and Gram-negative (*E. coli*, *S. marcescens*) bacteria tested (Figure 14a–j). These results are in concordance with other reported studies [80,81]. Microcrystalline cellulose disks of 1.4 cm diameter did not exhibited antibacterial activity against *E. coli* and *S. aureus* [80]. In another study, nanofibrillar cellulose films of 9 cm diameter from eucalyptus pulp (NAP9) did not present antibacterial activity against *E. coli* and *S. aureus* [81].

## 3. Conclusions

Two types of hydrogels based on vegetal nanocellulose from brewer’s spent grains (VNC) and, respectively, bacterial nanocellulose from kombucha fermentation (BNC) were obtained. Morphological analyses showed a denser fibrillar structure with longer cellulose nanofibers in the case of BNC compared to VNC. BNC showed higher crystallinity and hydrophilicity than VNC. The mucin binding efficiency of the BNC hydrogel was significantly higher (92.85 ± 2.62% and 82.18 ± 2.22% at a nanocellulose/mucin ratio (mg/mg) of 12 and 4, respectively) in comparison with the VNC hydrogel (74.45 ± 2.54% and 46.66 ± 1.47% at a nanocellulose/mucin ratio (mg/mg) of 12 and 4, respectively). The interaction between BNC and mucin was stronger compared to the interaction between VNC and mucin based on rheological analysis.

The BNC hydrogel presented a high degree of cytocompatibility for all the concentrations tested. At 0.025% (*w*/*v*) BNC hydrogel, the highest increase in the number of metabolically active viable cells, i.e., 107.60 ± 0.98% of cytotoxicity negative control, was recorded, and also a decrease in the amount of ROS with approx. 12% in comparison with the positive control. The lowest ROS production was recorded following the VNC hydrogel treatment, with an approx. 23% decrease in the ROS level in comparison with the positive control. None of the hydrogels showed antibacterial activity. This suggests that the BNC hydrogel represents, from some points of view, an excellent and better than VNC candidate for various biomedical nanoformulations. Nevertheless, antioxidant and antimicrobial characteristics of the developed formulations need to be additionally improved. Kombucha SCOBY should be considered as a cost-effective source of bacterial cellulose, but new, more environment-friendly purification processes should be developed.

## 4. Materials and Methods

### 4.1. Materials

Brewer’s spent grains were recovered from the Ursus brewery (Brasov, Romania) after the beer production process. The symbiotic culture of bacteria and yeasts (SCOBY) pellicles were obtained from a local kombucha culture [29].

The black tea (Twinings of London, Andover, Hampshire, UK) and the sucrose (commercial white crystalline sugar, Mărgăritar, Agrana, Wien, Austria) were used in order to prepare the sweetened tea infusion. Sodium hydroxide pellets and sodium hypochlorite solution 15% *w*/*v* (Scharlau, Barcelona, Spain) were used in order to purify the vegetal and bacterial cellulose.

The mucin binding efficiency was investigated using the following reagents: fuchsin basic for microscopy, hydrochloric acid 1N (Scharlau, Barcelona, Spain), glacial acetic acid, potassium metabisulfite, activated charcoal (Chimreactiv, Bucharest, Romania), mucin from porcine stomach type II (Sigma-Aldrich, St. Louis, MO, USA), and periodic acid (VWR International, Radnor, PA, USA).

The antioxidant activity of the VNC and BNC hydrogels was assessed using the following reagents: 2,2-diphenyl-1-picrylhydrazyl (DPPH), ethanol 96% (Chimreactiv SRL, Bucharest, Romania), Trolox 97% (Acros Organics, Thermo Fisher Scientific, Pittsburgh, PA, USA), potassium ferricyanide (Reactivul, Bucharest, Romania), L(+)-ascorbic acid, trichloroacetic acid, disodium hydrogen phosphate dihydrate, sodium dihydrogen phosphate monohydrate (Scharlau, Barcelona, Spain), and iron chloride (III) (VWR, Leuven, Belgium).

The investigation of cell viability was performed on human gingival fibroblasts (HGF-1, ATCC CRL-2014) using Cell Counting Kit-8 (Bimake, Houston, TX, USA) and the Viability/Cytotoxicity Assay Kit (Biotium, Fremont, CA, USA). The cell morphology was highlighted using Phalloidin-iFluor 488 Reagent (Abcam, Cambridge, United Kingdom) and 4′,6-diamidino-2-phenyindole, dilactate (DAPI) (Sigma-Aldrich, St. Louis, MO, USA). In order to investigate the cytocompatibility, different buffers or culture media were prepared based on Dulbecco’s Modified Eagle’s Medium (DMEM)-low glucose, D-(+)-Glucose, sodium bicarbonate, trypsin from porcine pancreas, dimethyl sulfoxide 99.5%, antibiotic antimycotic solution 100× stabilized, trypsin from porcine pancreas (Sigma-Aldrich, St. Louis, MO, USA), di-sodium hydrogen phosphate dihydrate, sodium dihydrogen phosphate monohydrate, potassium chloride, sodium chloride, paraformaldehyde, Triton X-100 (Scharlau, Barcelona, Spain), albumin bovine fraction V, pH 7.0 (Janssen Chimica, Beerse, Belgium), and fetal bovine serum (FBS) USDA APPD. ORIGIN (Thermo Fisher Scientific, Waltham, MA, USA). For the assessment of the in vitro antioxidant activity, the following reagents were used: 2,7-dichlorodihydrofluorescein (Cayman Chemicals, Ann Arbor, MI, USA), dilactate sodium deoxycholate, ethylenediaminetetraacetic acid tetrasodium salt dihydrate, hydrochloric acid 37% (Sigma Aldrich, St. Louis, MO, USA), Triton X-100, sodium n-dodecyl sulfate 99%, 30% hydrogen peroxide, and tris-(hydroxymethyl)-aminomethane (Scharlau, Barcelona, Spain).

The following bacterial strains were used in order to investigate the antibacterial activity of VNC and BNC hydrogels: *Bacillus cereus* NCTC 10320, *Enterococcus faecalis* ATCC 29212, *Staphylococcus aureus* ATCC 25923, *Escherichia coli* ATCC 25922, and *Serratia marcescens* NCTC 10211. For the semi-quantitative screening of the antibacterial activity, different culture media or buffers were prepared based on Müeller–Hinton broth (MHB), Müeller–Hinton agar (MHA), and sodium chloride (Scharlau, Barcelona, Spain).

### 4.2. Production of Bacterial Cellulose

Bacterial cellulose: the synthesis of bacterial cellulose was induced according to [46]. A tea infusion of 10 g/L black tea in sterile double-distilled water was prepared (5 min at 95 °C), then filtered through sterile gauze. The sweetened tea infusion was obtained by adding 80 g/L sucrose (commercial white crystalline sugar). After the sweetened tea infusion reached room temperature, the fermentation process was started by adding 10% SCOBY from a previous kombucha fermented beverage. The jars were covered with sterile gauze and left at room temperature for 14 days.

### 4.3. Production of Bacterial and Vegetal Cellulose Nanofibers

Due to the recalcitrant vegetable biomass, a first stage of purification was necessary in the case of BSG. BSG was ground with a Retsch Type S 100 centrifugal mill (Haan, Germany). In order to remove the extractables, BSG was mixed with a solution of 2:1 toluene:ethanol (*v*/*v*) in a ratio of 1:20 (*w*/*v*) and kept for 6 h at 90 °C, 250 rpm, in a laboratory reflux installation. Subsequently, the resulting precipitate was centrifuged in a Universal 320R Centrifuge (Hettich, Tuttlingen, Germany) at 12 °C, 7350× *g* for 30 min and freeze-dried for 24 h using a ScanVac CoolSafe 55-4 freeze-dryer (LaboGene, Bjarkesvej, Denmark) at a working temperature −55 °C.

In order to obtain the vegetal and bacterial nanocellulose, the bacterial cellulose and pretreated BSG underwent a chemical purification process and a mechanical microfluidization process. Kombucha membranes were rinsed with double-distilled water and gently chopped before starting the treatments.

The purification process was performed according to [82]. Kombucha membranes and pretreated BSG were mixed with a solution of 1M NaOH at a ratio of 2.5 (*w*/*v*) and kept at a temperature of 90 °C for 1 h under stirring (250 rpm). The alkaline solution was removed, and the samples were thoroughly washed with double-distilled water through centrifugation (7350× *g*, 30 min, room temperature). The cellulose samples were centrifuged as previously mentioned because when we tried to wash them with double-distilled water by using a 1 mm diameter sieve, the vegetal cellulose partially passed through it. The alkaline treatment was repeated, keeping the same parameters (90 °C, 1 h, 250 rpm), as well as the washing process. The washed samples were mixed with a solution of 1.5% NaClO and kept for 2 h at room temperature under stirring (250 rpm), then washed very well with double-distilled water. For a more effective purification, four ultrasound cycles of 6 h each were applied on the washed samples in water with an ultrasound bath Elmasonic P30H (Singen, Germany) using the following parameters: 60 °C, 37 pulses/min, and 100 W power.

The purified samples were subjected to a milling process using a blender. Suspensions of 6% wet substance (0.06% dry weight) were prepared with double-distilled water and underwent a mechanical process of 20 passes using a microfluidizer (Microfluidics, Westwood, MA, USA). The dry weight of the samples was determined gravimetrically by drying a known volume of sample at 105 °C in an FD-S 115 drying and heating chamber (Binder, Tuttlingen, Germany) until a constant weight was reached.

### 4.4. Preparation of VNC-/BNC Hydrogels, and Hydrogel–Mucin Systems

Nanocellulose from brewer’s spent grains, as well as from kombucha fermentation synthetized in Section 4.3, were concentrated in a Rotavapor R-300 system (BUCHI Corporation, New Castle, DE, USA) up to a concentration of 0.4% dry weight using the following parameters: 60 °C evaporation temperature with a chiller stage at 20 °C, 40 rpm, and 100 bar pressure, which resulted in VNC and BNC samples. Explicitly, from 1 L of 0.06% (dry weight) nanocellulose suspension, we obtained a volume of 0.15 L that led to the formation of a gel-like suspension. The dry weight of the samples was determined as in Section 4.3.

In order to assess the physicochemical properties of the hydrogel–mucin system, a suspension of 3.5% mucin was prepared in double-distilled water. After the complete homogenization of the suspension according to [50], it was mixed with VNC- and BNC hydrogels at a ratio of 1:1 (*v*/*v*), which resulted in VNCMu and BNCMu samples.

For the analyses where it is mentioned that the samples were freeze-dried, the same parameters mentioned in Section 4.3 were used for the freeze-drying process of the VNC, BNC, VNC-Mu, BNC-Mu, and Mu samples. When the analyses were performed on non-freeze-dried samples, it is mentioned that the samples were in their initial/original state.

### 4.5. Physico-Chemical Characterization of the VNC/BNC Hydrogels and Investigation of the Hydrogel–Mucin Interaction

#### 4.5.1. Transmission Electron Microscopy (TEM) Analysis

A small amount of samples in their initial/original state (10 µL) was added on a lacey formvar/carbon type-B film and 200 mesh copper grid (Ted Pella, Redding, CA, USA). TEM images were acquired with a TECNAI F20 G2 TWIN Cryo-TEM (FEI) transmission electron microscope (Houston, TX, USA) in bright-field mode, applying an accelerating voltage of 200 kV.

#### 4.5.2. Scanning Electron Microscopy (SEM) Analysis

SEM micrographs of freeze-dried samples were acquired with a TM4000Plus II tabletop electron microscope (Hitachi, Tokyo, Japan) using the following parameters: 15 kV electron acceleration voltage, secondary electrons (SE)/backscattered electrons (BSE) detector, low-charge (L) vacuum mode, and 1000× magnification.

#### 4.5.3. X-Ray Diffraction (XRD) Analysis

The XRDs were performed on freeze-dried samples with a Rigaku–SmartLab diffractometer (Rigaku Corporation, Tokyo, Japan) using the following parameters: 40 kV operation voltage, 200 mA emission current, and incident Cu_Kα1_ radiation (λ = 1.54059 Å). A range of 2θ angles 5–50° with a step of 0.02° and a scan speed of 4°/min was used in order to obtain the diffractograms. The diffractograms were processed and analyzed in the PDXL software version 2.7.2.0 for the peak identification and calculation of the crystallinity degree (Xc, %), which is the ratio between the area of the crystalline peaks and the area of all peaks.

#### 4.5.4. Interfacial Tension and Contact Angle Assessment

An OCA50 optical tensiometer (DataPhysics Instruments GmbH, Filderstadt, Germany) was used in order to measure the surface tension (mN/m) and the contact angle (°) of the samples in their initial/original state (0.4% nanocellulose suspension as described in Section 4.4). The measurements were performed in triplicate for each sample using the SCA20 software version 5.0.41.

#### 4.5.5. Investigation of Mucin Binding Efficiency by Periodic acid Schiff (PAS) Assay

The VNC- and BNC hydrogels in their initial/original state (0.4% dry matter) were mixed with a suspension of 0.1% mucin prepared in double distilled water (*w*/*v*) at a 3:1 and 1:1 (*v*/*v*) ratio, then incubated at 37 °C (Static Cooled incubator MIR-154 PHCbi, Tokyo, Japan) under shaking (Trayster IKA, Staufenim Breisgau, Germany) for 1 h. The samples were centrifuged for 1 h at room temperature, 20,000× *g*. Free mucin in the supernatant was quantified by the PAS reaction [83,84]. The mucin binding efficiency (%) was calculated by subtracting the free mucin concentration from the initial mucin concentration.

#### 4.5.6. Rheological Analysis

Rheological experiments of VNC and BNC hydrogels in their initial/original state, together with the hydrogel–mucin systems, were performed using a HR20 Discovery Hybrid rotational rheometer from TA Instruments (New Castle, DE, USA) in three different shearing modes with hysteresis (up and reverse rate variation) at 25 °C and 500 µm gap. A sample amount of approx. 0.7 mL, able to fill the 500 µm gap between the 40 mm geometry and quartz surface, was subjected to a series of UP and reverse (R) oscillation mode in the angular frequency range of ω 0.1–100 rad/s, followed by a UP and R linear flow sweep in the shear rate range of 0.1–100 1/s, and ending with the axial mode of geometry rising with a constant speed of 10 µm/s for a duration of 10 min to determine the adhesion force and adhesion time. The rheological behavior was mathematically evaluated by regression of the viscosity and stress curves in flow sweep mode with the available rheological models in the Trios software version 5.1.1 (TA Instruments-Waters LLC, New Castle, DE, USA). The models Cross, Carreau, Carreau–Yasuda, Sisko, and Williamson were applied for viscosity curves and compared using the coefficient of determination R^2^. The models Newtonian, Bingham, Casson, power-law (Ostwald de Waele), and Herschel–Bulkley were applied for stress curves, as in a previous study [50]. Additionally, the thixotropic index was calculated for the stress curves as the area between the up and reverse curves. The normalized thixotropy is defined in the Trios software as the thixotropic index divided by the maximum rate, the maximum stress, and the step time.

#### 4.5.7. Antioxidant Activity Assessment by DPPH and Potassium Ferricyanide Reducing Power (PFRAP) Assays

For the DPPH assay, 300 μL of the samples in their initial/original state were mixed with 300 μL of 300 μM DPPH reagent in 96% ethanol. After 30 min of incubation at room temperature, the samples were centrifuged at 6000× *g*, and the supernatant was pipetted in a new plate. The absorbance was read at λ = 517 nm. The calibration curve was prepared starting from a stock solution of 1 mM Trolox in 70% ethanol [50]. The results are expressed as µM Trolox equivalents (TE)/mg hydrogel (dry weight).

For the PFRAP assay, 125 µL of samples in their initial/original state were mixed with 250 µL of 0.2 M sodium phosphate buffer (pH 6.6) and 250 µL of 1% K_3_Fe(CN)_6_. The samples were incubated for 20 min at 50 °C. Afterwards, 250 µL of 10% trichloroacetic acid was added in the reaction mixture and the samples were centrifuged at 6000× *g*. The resulting supernatant was mixed with 250 µL of double-distilled water and 50 µL of 0.1% FeCl_3_. The absorbance was read at 700 nm. The calibration curve was prepared starting from a stock solution of 1 mg/mL ascorbic acid [85]. The results are expressed as µM ascorbic acid equivalents (AAE)/mg hydrogel (dry weight).

#### 4.5.8. Fourier Transform Infrared Spectroscopy Analysis

The ATR-FTIR determinations of freeze-dried samples were performed with an IRTracer-100 spectrophotometer (Shimadzu, Kyoto, Japan) equipped with a total reflection attenuation accessory. Each spectrum was the mean of 45 scans with a resolution of 4 cm^−1^ in the mid-IR spectral range of 4000–400 cm^−1^.

#### 4.5.9. Porosity Analysis

The porosity and associated morphological properties of freeze-dried VNC and BNC were analyzed by nitrogen adsorption–desorption analyses using the Autosorb Quantachrome Nova 2200e Analyzer (Boynton Beach, FL, USA). The samples were firstly degassed for 20 h under a vacuum at 105 °C, weighted before and after degassing, and measured at 77 K using liquid nitrogen. The VNC sample was proved to be unsuitable for this analysis, even after duplicate analysis, probably because of the very low porosity. For BNC, the software Quantachrome NovaWin version 11.03 was used to determine the specific surface area, specific pore volume, pore dimensions (micro-, meso-, or macropores), and pore size distribution. Two build-in models were compared, respectively, Brunauer–Emmett–Teller (BET) and quenched solid density functional theory (QS-DFT), using data reduction parameters nitrogen adsorbate at 77 K, carbon adsorbent with slit and cylindrical pores, thermal transpiration ON, and ACARB standard isotherm file.

### 4.6. Biological Activity of VNC and BNC

#### 4.6.1. Cytocompatibility Analysis by Cell Counting Kit-8 (CCK-8) and LIVE/DEAD Assays

For the investigation of cytocompatibility of nanocellulose-based hydrogels, DMEM was supplemented with 10% FBS. HGF-1 cells were seeded in 48-well plates using a cell density of 1 × 10^4^ cells/cm^2^ and kept at 37 °C under 5% CO_2_ atmosphere. At 24 h post-seeding, the cells were treated with different concentrations of VNC and BNC hydrogels in their initial/original state, i.e., 0.0125, 0.025, 0.05, 0.1, and 0.2% dry weight (*w*/*v*). Cell viability was determined 24 h post-treatment by combining the CCK-8 and LIVE/DEAD assays according to [28]. In the case of CCK-8 assay, after the incubation time, the liquid was transferred in a 96-well plate and the absorbance was recorded at 450 nm using a microplate reader (CLARIOstar BMG Labtech, Ortenberg, Germany). After performing the LIVE/DEAD assay, HGF-1 cells were visualized using a cell imaging system (Celena^®^ X High Content Imaging System (Logos Biosystems, Gyeonggi-do, South Korea)). The image acquisition was performed using Celena^®^ X Explorer software version 1.0.5 (4× objective), and the image analysis was carried out using Celena^®^ X Cell Analyzer version 1.5.2.

#### 4.6.2. Investigation of Cell Morphology

The morphological features of the cells 24 h post-treatment (as in Section 4.6.1) were highlighted by fluorescent labeling of cytoskeletal actin with Alexa Fluor 488-coupled phalloidin (green) and nuclei with DAPI (blue) according to [28]. The HGF-1 cells were examined using the Celena^®^ X High Content Imaging System (Logos Biosystem, Gyeonggi-do). The image acquisition was carried out using Celena^®^ X Explorer software, 1.0.5 (20× objective). The image processing (overlapping fluorescent images of cytoskeletal actin and nuclei) was made using the Celena^®^ X Cell Analyzer, version 1.5.2.

#### 4.6.3. In Vitro Antioxidant Activity

The same cell density and growth conditions described in Section 4.6.1 were used for the investigation of the in vitro antioxidant activity. The HGF-1 cells were incubated in the presence of the VNC or BNC hydrogel in its initial/original state (0.025% dry weight) and a solution of 37 μM H_2_O_2_ (ROS inducer). The samples were followed by a negative control (untreated cells) and a positive control (cells treated with the ROS inducer). For the total intracellular ROS labeling (qualitative analysis), the cells were washed with serum-free medium and incubated for 30 min with a solution of 10 μM 2′,7′- H_2_DCFDA which was prepared according to [46]. Afterwards, the cells were washed twice with serum-free medium. After a final wash with phosphate saline buffer (PBS), the fluorescence microscopy images were acquired using Celena^®^ X Explorer software, 1.0.5 (4× objective), and the image analysis was carried out using Celena^®^ X Cell Analyzer version 1.5.2. For the quantitative analysis, the PBS was replaced with the lysis buffer (RIPA) and the 48-well plate was transferred on ice for 5 min. The cell lysate was centrifuged 10 min at 19,000× *g*. The fluorescence intensity was measured using a microplate reader (485 nm excitation, 530 nm emission) [86].

#### 4.6.4. Antibacterial Activity

The diffusimetric method was used for the semi-quantitative screening of the antimicrobial activity of the hydrogel samples in their initial/original state (0.4% nanocellulose dry weight). The bacterial strains were activated by being cultivated for 24 h in MHB media at 37 °C. Subsequently, the activated bacterial strains were transferred onto MHA media, which was previously poured into 90 mm Petri dishes. The plates were subjected to the same conditions (24 h, 37 °C incubation) previously mentioned. After the 24 h of incubation, the standardized bacterial inoculum of 0.5 McFarland was prepared in sterile physiological water (0.8% NaCl) [50].

The first antimicrobial screening involved the droplet distribution of 10 μL of VNC and BNC hydrogels on MHA media previously seeded with the standardized bacterial suspension. The second antimicrobial screening implied a 6 mm diameter well made with a cork-borer in the center of the MHA media previously seeded with the standardized bacterial suspension. Afterwards, 50 µL of the VNC or BNC hydrogels were added in the well [87]. At 24 h post-incubation, the plates were photographed.

### 4.7. Statistical Analysis

Statistical analysis (One-Way ANOVA, Independent Sample T-test) was performed using IBM SPSS 26 software version 26.0.0.0 (IBM Corp., Armonk, NY, USA).

## Figures and Tables

**Figure 1 gels-11-00037-f001:**
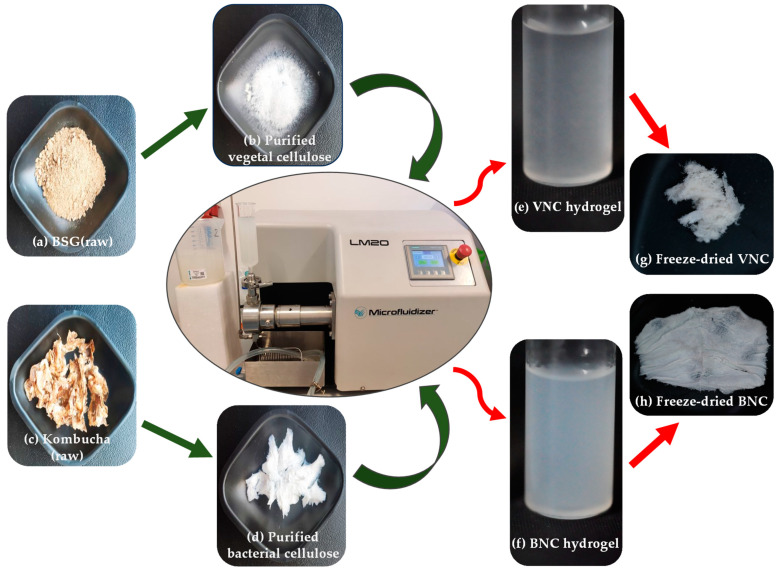
Schematic representation of cellulose purification and hydrogel preparation: (**a**) brewer’s spent grains (BSG); (**b**) purified vegetal cellulose from BSG; (**c**) bacterial cellulose (BC) membrane from kombucha fermentation; (**d**) purified BC; (**e**) hydrogel of vegetal nanocellulose from BSG (VNC); (**f**) hydrogel of bacterial nanocellulose from kombucha fermentation (BNC); (**g**) freeze-dried VNC; and (**h**) freeze-dried BNC.

**Figure 2 gels-11-00037-f002:**
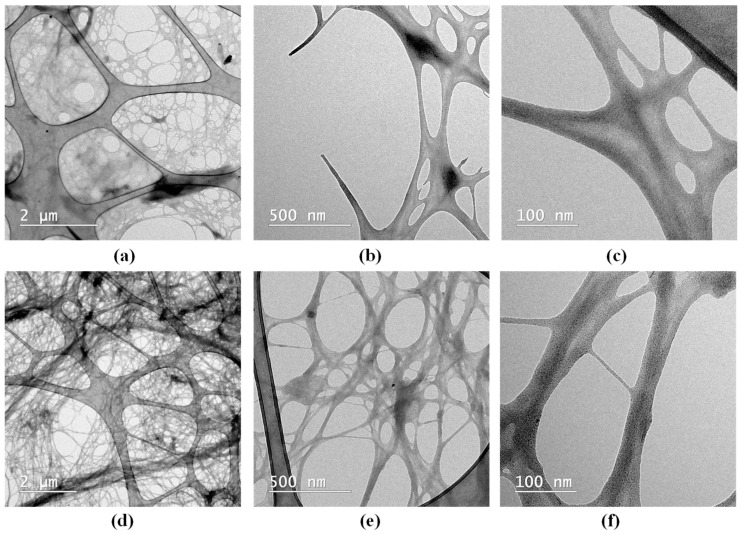
TEM analysis of: (**a**) VNC (2 µm scale); (**b**) VNC (500 nm scale), (**c**) VNC (100 nm scale); (**d**) BNC (2 µm scale); (**e**) BNC (500 nm scale); and (**f**) BNC (100 nm scale); VNC—hydrogel of vegetal nanocellulose from brewer’s spent grains; and BNC—hydrogel of bacterial nanocellulose from kombucha fermentation.

**Figure 3 gels-11-00037-f003:**
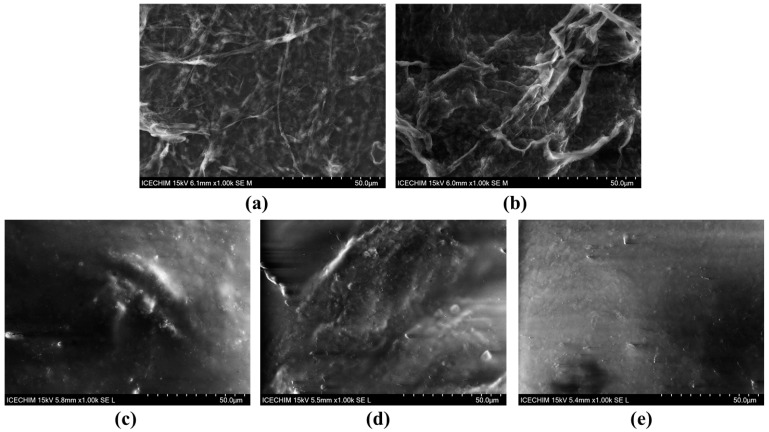
SEM analysis using secondary electrons (SE) detector (1000×) of: (**a**) VNC, (**b**) BNC, (**c**) VNCMu, (**d**) BNCMu, and (**e**) Mu; VNC—hydrogel of vegetal nanocellulose from brewer’s spent grains-based hydrogel; BNC—hydrogel of bacterial nanocellulose from kombucha fermentation-based hydrogel; VNCMu—VNC mixed with a 3.5% mucin suspension in a ratio of 1:1 (*v*/*v*); BNCMu—BNC mixed with a 3.5% mucin suspension in a ratio of 1:1 (*v*/*v*); and Mu—mucin suspension.

**Figure 4 gels-11-00037-f004:**
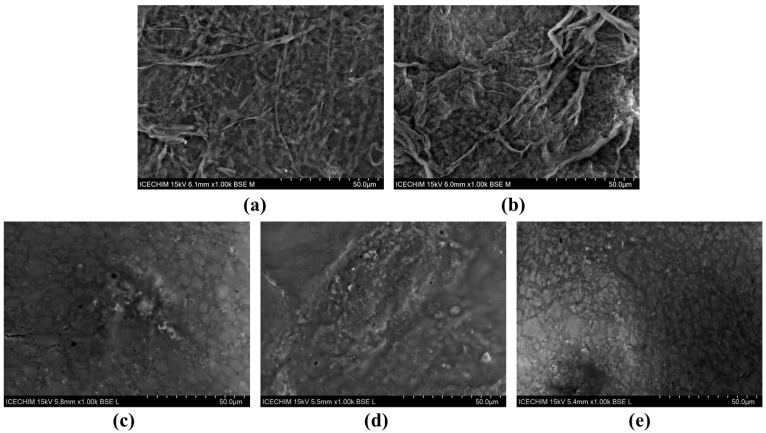
SEM analysis using backscattered electrons (BSE) detector (1000×) of: (**a**) VNC, (**b**) BNC, (**c**) VNCMu, (**d**) BNCMu, (**e**) Mu; VNC—hydrogel of vegetal nanocellulose from brewer’s spent grains; BNC—hydrogel of bacterial nanocellulose from kombucha fermentation; VNCMu—VNC mixed with a 3.5% mucin suspension in a ratio of 1:1 (*v*/*v*); BNCMu—BNC mixed with a 3.5% mucin suspension in a ratio of 1:1 (*v*/*v*); and Mu—mucin suspension.

**Figure 5 gels-11-00037-f005:**
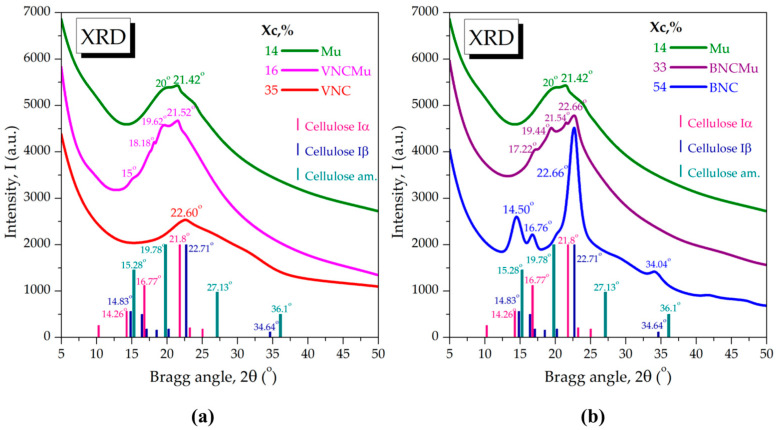
X-ray diffraction (XRD) analysis and crystallinity index (Xc,%) of: (**a**) VNC, Mu, VNCMu; (**b**) BNC, Mu, BNCMu; the vertical bars represent the main diffraction peaks of cellulose Iα, Iβ, and amorphous cellulose in the PDXL database; VNC—hydrogel of vegetal nanocellulose from brewer’s spent grains; BNC—hydrogel of bacterial nanocellulose from kombucha fermentation; VNCMu—VNC mixed with a 3.5% mucin suspension in a ratio of 1:1 (*v*/*v*); BNCMu—BNC mixed with a 3.5% mucin suspension in a ratio of 1:1 (*v*/*v*); and Mu—mucin suspension.

**Figure 6 gels-11-00037-f006:**
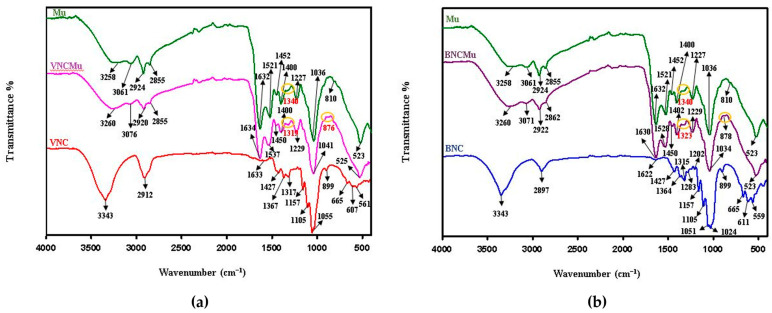
Overlapping ATR-FTIR spectra for (**a**) VNC, Mu, VNCMu, and (**b**) BNC, Mu, and BNCMu. VNCMu—hydrogel of vegetal nanocellulose from brewer’s spent grains (VNC) mixed with a mucin suspension in a ratio of 1:1 (*v*/*v*); and BNCMu—hydrogel of bacterial nanocellulose from kombucha fermentation (BNC) mixed with a mucin suspension in a ratio of 1:1 (*v*/*v*).

**Figure 7 gels-11-00037-f007:**
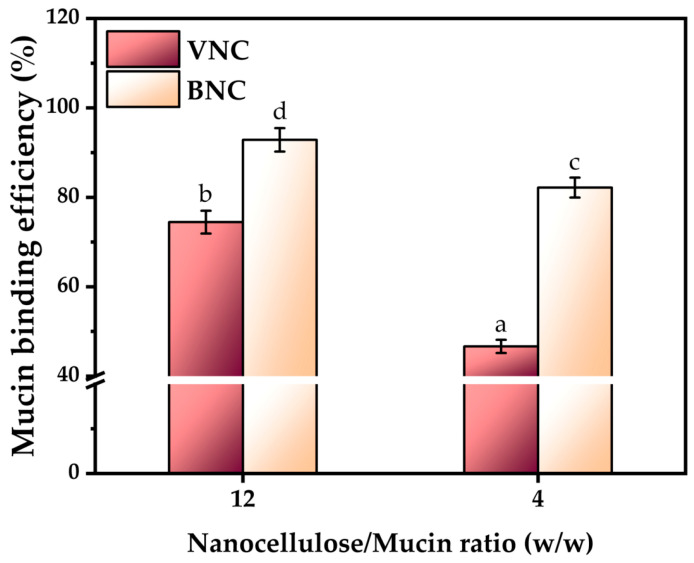
Mucin binding efficiency (±standard deviation, *n* = 3, α < 0.05; different letters indicate statistically significant differences between samples); VNC—hydrogel of vegetal nanocellulose from brewer’s spent grains; and BNC—hydrogel of bacterial nanocellulose from kombucha fermentation.

**Figure 8 gels-11-00037-f008:**
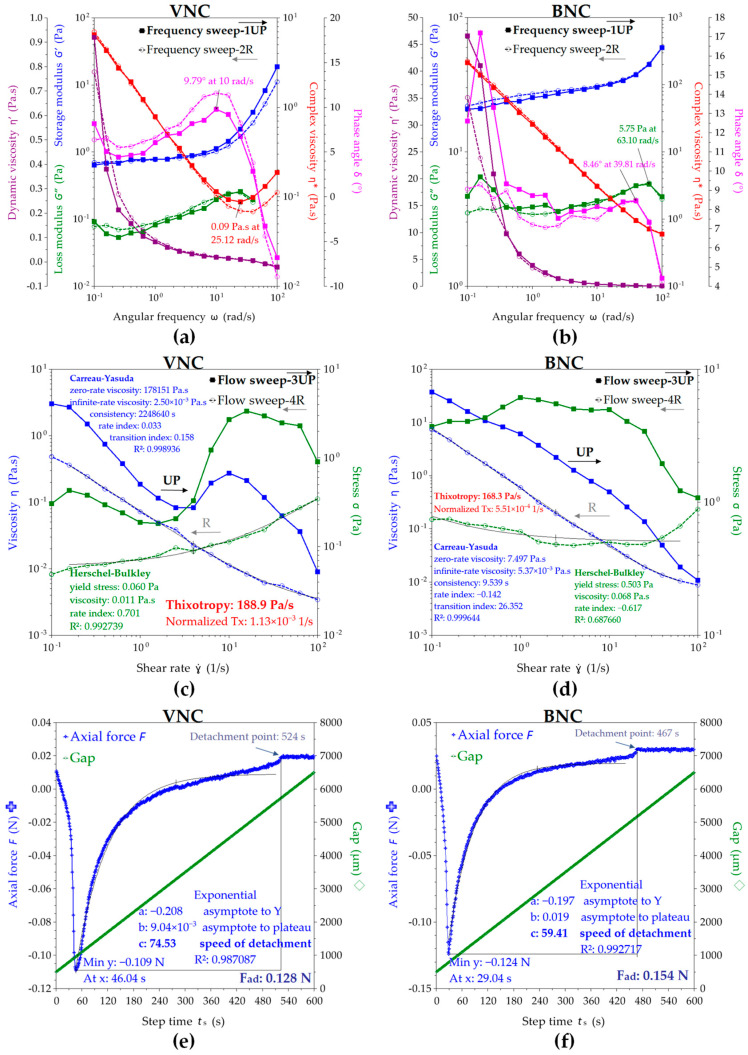
Rheological behavior of VNC and BNC hydrogels: (**a**) frequency sweep of VNC; (**b**) frequency sweep of BNC; (**c**) flow sweep of VNC; (**d**) flow sweep of BNC; (**e**) axial mode of VNC; (**f**) axial mode of BNC; VNC—hydrogel of vegetal nanocellulose from brewer’s spent grains; and BNC—hydrogel of bacterial nanocellulose from kombucha fermentation.

**Figure 9 gels-11-00037-f009:**
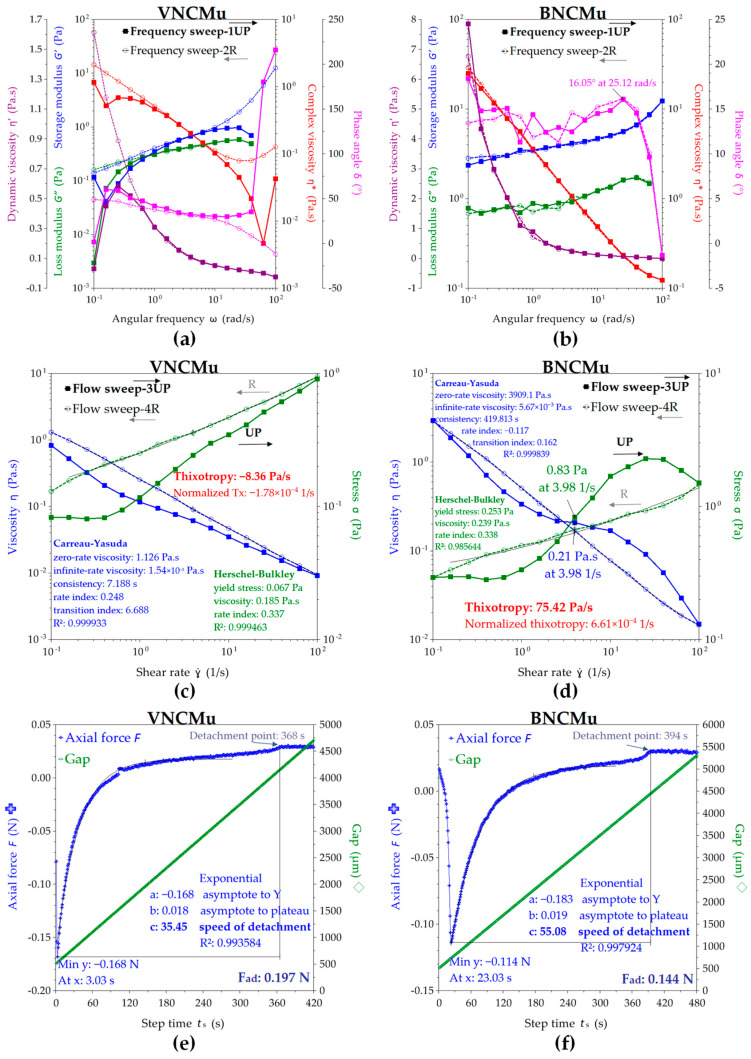
Rheological behavior of VNCMu and BNCMu: (**a**) frequency sweep of VNCMu; (**b**) frequency sweep of BNCMu; (**c**) flow sweep of VNCMu; (**d**) flow sweep of BNCMu; (**e**) axial mode of VNCMu; (**f**) axial mode of BNCMu; VNCMu—hydrogel of vegetal nanocellulose from brewer’s spent grains (VNC) mixed with a mucin suspension in a ratio of 1:1 (*v*/*v*); and BNCMu—hydrogel of bacterial nanocellulose from kombucha fermentation (BNC) mixed with a mucin suspension in a ratio of 1:1 (*v*/*v*).

**Figure 10 gels-11-00037-f010:**
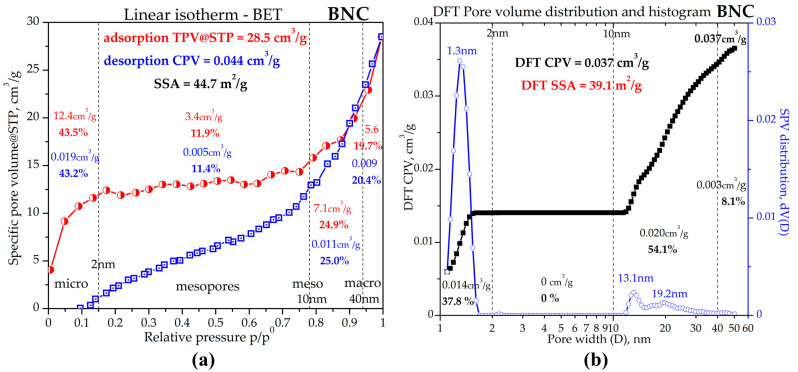
Porosity analysis of freeze-dried BNC: (**a**) BET isotherm and pore volume distribution for micropores (D < 2 nm), small mesopores (2 < D < 10 nm), large mesopores (10 < D < 40 nm), and macropores (D > 40 nm); and (**b**) DFT method for cumulative pore volume and pore size distribution. BNC—hydrogel of bacterial nanocellulose from kombucha fermentation.

**Figure 11 gels-11-00037-f011:**
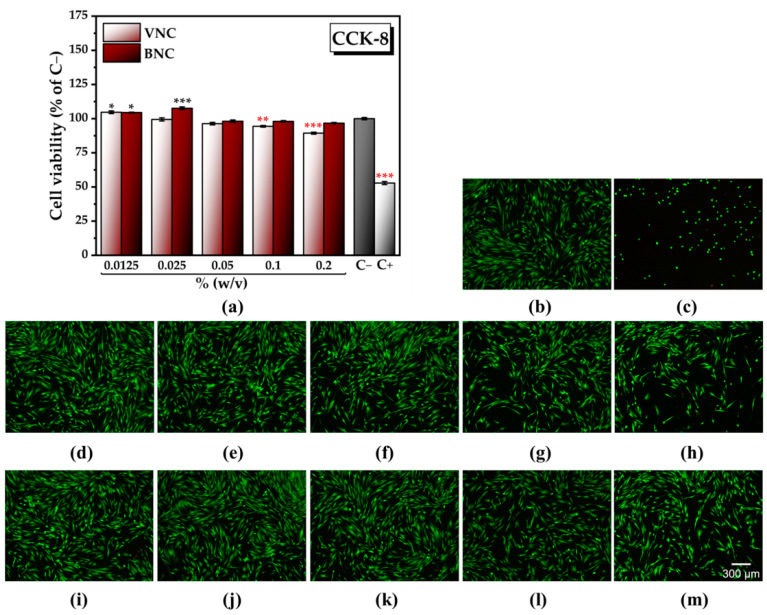
Cytocompatibility of VNC and BNC hydrogels: (**a**) Cell Counting Kit-8 (CCK-8) assay (±error bars, *n* = 3, α < 0.05; *—σ between 0.05 and 0.01, **—σ between 0.01 and 0.001, and ***—σ < 0.001; black stars indicate statistically significant values that exceed C−; red stars indicate statistically significant values that are below C−); C− (untreated cells, negative cytotoxicity control), C+ (cells treated with 7.5% dimethyl sulfoxide (DMSO), positive cytotoxicity control), VNC—hydrogel of vegetal nanocellulose from brewer’s spent grains; BNC—hydrogel of bacterial nanocellulose from kombucha fermentation; (**b**–**h**) LIVE/DEAD assay (live cells—green fluorescence, dead cells—red fluorescence): (**b**) C−; (**c**) C+; (**d**) cells treated with 0.0125% (*w*/*v*) VNC; (**e**) cells treated with 0.025% (*w*/*v*) VNC; (**f**) cells treated with 0.05% (*w*/*v*) VNC; (**g**) cells treated with 0.1% (*w*/*v*) VNC; (**h**) cells treated with 0.2% (*w*/*v*) VNC; (**i**) cells treated with 0.0125% (*w*/*v*) BNC; (**j**) cells treated with 0.025% (*w*/*v*) BNC; (**k**) cells treated with 0.05% (*w*/*v*) BNC; (**l**) cells treated with 0.1% (*w*/*v*) BNC; and (**m**) cells treated with 0.2% (*w*/*v*) BNC.

**Figure 12 gels-11-00037-f012:**
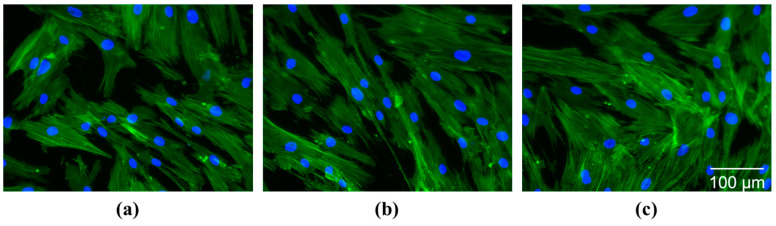
Cell morphology following the treatment with VNC and BNC hydrogels (Alexa Fluor 488-coupled phalloidin labelling of the actin filaments—green fluorescence, and DAPI labelling of the nuclei—blue fluorescence): (**a**) untreated cells, negative control); (**b**) cells treated with 0.025% (*w*/*v*) VNC; (**c**) cells treated with 0.025% BNC; VNC—hydrogel of vegetal nanocellulose from brewer’s spent grains; and BNC—hydrogel of bacterial nanocellulose from kombucha fermentation.

**Figure 13 gels-11-00037-f013:**
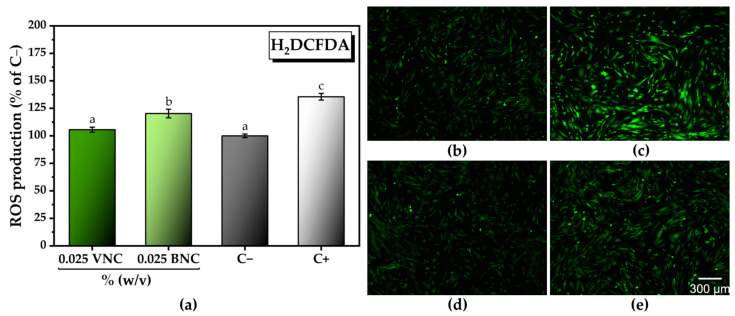
In vitro antioxidant activity following the treatment with VNC and BNC hydrogels: (**a**) total intracellular reactive oxygen species (ROS) production (±standard deviation, *n* = 3, α < 0.05; different letters indicate statistically significant differences between samples); (**b–e**) fluorescence microscopy images after labeling total intracellular ROS with H_2_DCFDA (green fluorescence); HGF-1 cells treated with: (**b**) untreated cells (negative control); (**c**) cells treated with 37 µM H_2_O_2_ (positive control, ROS inducer) VNC; (**d**) cells treated with 0.025% (*w*/*v*) VNC in the presence of the ROS inducer; (**e**) cells treated with 0.025% BNC in the presence of the ROS inducer; VNC—hydrogel of vegetal nanocellulose from brewer’s spent grains; and BNC—hydrogel of bacterial nanocellulose from kombucha fermentation.

**Figure 14 gels-11-00037-f014:**
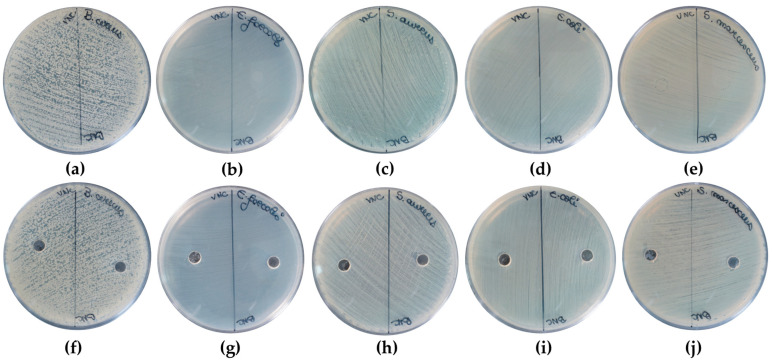
Antibacterial activity following the treatment with different VNC and BNC hydrogel doses: (**a**–**e**) antibacterial activity of 10 µL hydrogel dose against: (**a**) *B. cereus*; (**b**) *E. faecalis*; (**c**) *S. aureus*; (**d**) *E. coli*; (**e**) *S. marcescens*; (**f**–**j**) antibacterial activity of 50 µL hydrogel dose against: (**f**) *B. cereus*; (**g**) *E. faecalis*; (**h**) *S. aureus*; (**i**) *E. coli*; and (**j**) *S. marcescens*. VNC—hydrogel of vegetal nanocellulose from brewer’s spent grains; and BNC—hydrogel of bacterial nanocellulose from kombucha fermentation.

**Table 1 gels-11-00037-t001:** Surface tension and contact angles of nanocellulose-based hydrogels (VNC and BNC) and hydrogel–mucin systems (VNCMu and BNCMu) ± standard error (*n* = 3, α < 0.05).

Sample	Surface Tension (mN/m)	Contact Angle/Hydrophilic Surface (°)	Contact Angle/Hydrophobic Surface (°)
VNC *	50.44 ± 0.66, *d ***	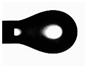	52.70 ± 0.56, *d*	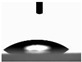	56.90 ± 0.40, *a*	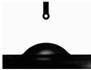
VNCMu	46.22 ± 0.39, *b*(8.37 ± 0.78% decrease)	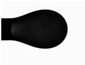	50.90 ± 0.03, *c*(3.42 ± 0.68% decrease)	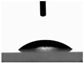	58.14 ± 0.41, *b*(2.18 ± 0.73% increase)	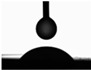
BNC	55.29 ± 0.29, *e*	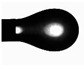	48.27 ± 0.57, *b*	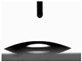	65.13 ± 0.35, *c*	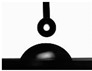
BNCMu	48.50 ± 0.54, *c*(12.28 ± 0.97% decrease)	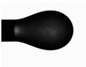	46.53 ± 0.55, *a*(3.59 ± 1.14% decrease)	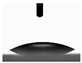	68.00 ± 0.26, *d*(4.40 ± 0.41% increase)	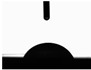
Mu	39.33 ± 0.47, *a*	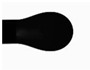	45.67 ± 0.71, *a*	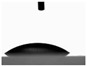	78.27 ± 0.45, *e*	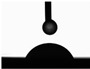

* VNC—hydrogel of vegetal nanocellulose from brewer’s spent grains; BNC—hydrogel of bacterial nanocellulose from kombucha fermentation; VNCMu—VNC mixed with a 3.5% mucin suspension with a ratio of 1:1 (*v*/*v*); BNCMu—BNC mixed with a 3.5% mucin suspension with a ratio of 1:1 (*v*/*v*); and Mu—mucin suspension. ** Different letters in italics indicate statistically significant differences between samples.

**Table 2 gels-11-00037-t002:** The antioxidant activity of VNC and BNC hydrogels.

Sample	DPPH (µM TE/mg (Dry Weight) Hydrogel *	PFRAP (µM AAE/mg (Dry Weight) Hydrogel
VNC **	13.6 ± 2.6	15.5 ± 1.2
BNC	0	0

* DPPH—2,2-diphenyl-1-picrylhydrazyl; TE—Trolox equivalents; PFRAP—potassium ferricyanide reducing power; and AAE—ascorbic acid equivalents. ** VNC—hydrogel of vegetal nanocellulose from brewer’s spent grains; and BNC—hydrogel of bacterial nanocellulose from kombucha fermentation.

## Data Availability

All data are presented within the article.

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
