# Peer review of "Kombucha Versus Vegetal Cellulose for Affordable Mucoadhesive (nano)Formulations"

_gels, 2025, doi:10.3390/gels11010037_

Round 1

Reviewer 1 Report

Comments and Suggestions for Authors

In an era of growing interest in bacterial cellulose, comparing its properties and potential at the physico-chemical level with the plant cellulose now most widely used is a topic worth investigating and describing in the scientific literature. This is what the authors of the paper ” Bacterial cellulose from Kombucha pellicle for mucoadhesive  nanoformulations – a comparison with vegetal cellulose from 3 brewer’s spent grains” write about

The question of the origin of the material is problematic. I know it is problematic in this situation but it affects the properties of the material dramatically and could change the result dramatically even if we treat it comparatively (bacterial cellulose vs plant). Maybe it would be worthwhile to obtain material from several breweries or from several manufacturing processes. As for kombucha, the problem of the inconstancy of the composition of the mixture of yeast and bacteria is a well-known issue. The author runs away from this problem by referring the reader to the literature which also does not specify this issue. This is undoubtedly a weak point of the publication. Instead of kombucha, one could use bacterial cellulose extracted from a specific bacterial strain from the microbial deposit. As for plant cellulose, there is unfortunately no simple solution. These issues should be discussed in the introduction.

Figure 2,3,4 auto repeatedly explains abbreviations. Once introduced an abbreviation should continue to be used without explanation.

The first subsection ‘2. Results and Discussion’ 2.1 on physico-chemical properties should still be divided into subsections . Without this, the paper reads badly when it all blends into one.

The description of the synthesis of bacterial cellulose ‘Production of bacterial cellulose’ has no information on which tea was used  and how it was brewed. The information that sugar was added to the medium without any information on what kind of “sugar” is unacceptable.

Surface tension measurements for freeze-dried hydrogel (or soaked?) do not seem obvious . The authors should describe how the experiment was carried out. The same with the contact angle.  There are only two lines on this topic in this coin.  The publication needs to be corrected before publication.

Author Response

In an era of growing interest in bacterial cellulose, comparing its properties and potential at the physico-chemical level with the plant cellulose now most widely used is a topic worth investigating and describing in the scientific literature. This is what the authors of the paper ”Bacterial cellulose from Kombucha pellicle for mucoadhesive nanoformulations – a comparison with vegetal cellulose from 3 brewer’s spent grains” write about

Point 1: The question of the origin of the material is problematic. I know it is problematic in this situation but it affects the properties of the material dramatically and could change the result dramatically even if we treat it comparatively (bacterial cellulose vs plant). Maybe it would be worthwhile to obtain material from several breweries or from several manufacturing processes. As for kombucha, the problem of the inconstancy of the composition of the mixture of yeast and bacteria is a well-known issue. The author runs away from this problem by referring the reader to the literature which also does not specify this issue. This is undoubtedly a weak point of the publication. Instead of kombucha, one could use bacterial cellulose extracted from a specific bacterial strain from the microbial deposit. As for plant cellulose, there is unfortunately no simple solution. These issues should be discussed in the introduction.

Response 1: Thank you for your observation. It is true that the vegetal cellulose could present some differences depending on the plant species. We selected brewer’s spent grain as this is one of the most available vegetal subproduct in large quantities, which allows and moreover requires valorisation at industrial scale (we included this at lines 106-108). Although we might expect some variations between breweries and manufacturing processes, we do not think it could majorly impact the properties of the extracted cellulose. In any case, this aspect could constitute the subject of a further, separate study, as the amount of information gathered would not fit in a single article. We believe that any start is better than no start. In the case of bacteria, we specifically selected kombucha because the cellulose membrane represents a byproduct in this case (of kombucha drink), whereas the bacterial cellulose from a specific bacterial strain is not a byproduct and is known to be more expensive due to the specific culture medium needed. Similar to the vegetal cellulose from BSG, the bacterial cellulose of kombucha could present some differences depending on the consortium, but this should not be an impediment for valorising this byproduct. Future works should and will probably shed light on these aspects. We have some comments on these aspects in the Introduction (lines 109-113). Previous studies have shown differences of nanocellulose even between single strains of the same species (https://doi.org/10.1016/j.coche.2019.04.005), therefore using individual strain does not completely solve the problem.

Point 2: Figure 2,3,4 auto repeatedly explains abbreviations. Once introduced an abbreviation should continue to be used without explanation.

Response 2: Thank you for the suggestion, that is logically. However, the MDPI publisher in its layout style guide (https://www.mdpi.com/authors/layout#_bookmark14) mentions in Section 3.5 Abbreviations the following: Note that the abstract, main text and figure/table/diagram captions are treated separately for abbreviations. […] The reason for this is that they are often displayed in isolation; for example, indexing services usually display only the abstract and you can browse the figures without the main text via the journal's website.” Although MDPI does not request specifically abbreviation definition in each Figure and Table, it is considered as a general rule that each table and figure must be a stand-alone body and understood independently of other references, for several reasons (easy understanding of the figure, other authors could use the graphical display in future works and need to have the definitions of the terms attached etc.). Therefore, we consider that the figure / table caption is an important part for the understanding and the interpretation of the scientific data by all the readers and that it should contain every time the explanation for each abbreviation in the figure / table.

Point 3: The first subsection ‘2. Results and Discussion’ 2.1 on physico-chemical properties should still be divided into subsections. Without this, the paper reads badly when it all blends into one.

Response 3: Thank you for the suggestion. We split the previous subsection 2.1 into subsection 2.1 related to the morphological and structural characterization and 2.2 related to the physicochemical properties.

Point 4: The description of the synthesis of bacterial cellulose ‘Production of bacterial cellulose’ has no information on which tea was used and how it was brewed. The information that sugar was added to the medium without any information on what kind of “sugar” is unacceptable.

Response 4: Thank you very much for your observation. We have included information regarding how the black tea infusion was prepared and also about the type of sugar used (sucrose – commercial white crystalline sugar). Also, in Section 4.1. Materials, we included information about the origin of the black tea and of the sucrose.

Point 5: Surface tension measurements for freeze-dried hydrogel (or soaked?) do not seem obvious. The authors should describe how the experiment was carried out. The same with the contact angle.  There are only two lines on this topic in this coin.  The publication needs to be corrected before publication.

Response 5: For the analyses conducted on freeze-dried samples, it is mentioned that the samples were freeze-dried. We added: “If the analyses were performed on non-freeze-dried samples, it is mentioned that the samples were in their initial / original state”. Therefore, we included in each subsection more information about this aspect. Thank you for all your valuable observations.

Reviewer 2 Report

Comments and Suggestions for Authors

In this manuscript, the authors report on the properties and cytocompatibility of hydrogels prepared from cellulose nanofibers derived from bacteria and plants. The present study is very interesting because it provides a basis for the future development of cellulose nanofiber-derived hydrogels, which have been the subject of many basic and applied studies. The authors' manuscript is well structured, interesting, and suggestive, but contains the following minor issues that could be improved.

1)        The comparison of the various properties of hydrogels in this study is very interesting. It will be useful for future research on cellulose nanofibers. However, the authors should consider the physical aspects of the hydrogels obtained and the correlation between the chemical structure and the experimental results. For example, are the chemical interactions in the hydrogels formed by cellulose nanofibers irrelevant to the present experimental results?

2)        It would help the reader understand this study better if the authors described how the appearance of the two cellulose nanofiber dispersions and hydrogels differed.

3)        Page 10, line 340, “The Carreau-Yasuda rheological model” and line 341, “Herschel-Bulkley fitting model”: The authors should provide the definitions and sources of these models and discuss their application to the present experimental results in light of previous studies.

4)        Figures 7 and 8: In Figures 7b, 7c, 8b, and 8c, the values are noted as thixotropy, but what do they mean? The authors should provide a definition and source for thixotropy in a single data and discuss these results.

Author Response

In this manuscript, the authors report on the properties and cytocompatibility of hydrogels prepared from cellulose nanofibers derived from bacteria and plants. The present study is very interesting because it provides a basis for the future development of cellulose nanofiber-derived hydrogels, which have been the subject of many basic and applied studies. The authors' manuscript is well structured, interesting, and suggestive, but contains the following minor issues that could be improved.

Point 1: The comparison of the various properties of hydrogels in this study is very interesting. It will be useful for future research on cellulose nanofibers. However, the authors should consider the physical aspects of the hydrogels obtained and the correlation between the chemical structure and the experimental results. For example, are the chemical interactions in the hydrogels formed by cellulose nanofibers irrelevant to the present experimental results?

Response 1: Thank you for the observation. We performed and added FTIR characterisation in order to provide more information on the chemical composition and interactions of the two types of nanocellulose forming the hydrogels. The chemical composition and chemical structure seem to be similar, at least as shown by FTIR analysis. Nevertheless, there are some small differences which can partially explain the properties. We included some explanations related to the correlation between the hydrogen bonds and free -OH available, hydrophilicity, and mucin binding (lines 401-407), as well as how it influences the cytocompatibility (lines 640-642). Additionally, we performed AOA (DPPH, PFRAP and in vitro ROS reduction in cells) and showed that VNC has higher AOA activity which we proposed could be caused by a higher presence of antioxidant impurities (polyphenols) in VNC (lines 567-583, Table 2, lines 643-662, Figure 13).

Point 2: It would help the reader understand this study better if the authors described how the appearance of the two cellulose nanofiber dispersions and hydrogels differed.

Response 2: Thank you for your suggestion. We added more information (Section 2.1, lines 124-129). We also included in Section 4.4. how we prepared the hydrogels: “Explicitly, from 1 L of 0.06% (dry weight) nanocellulose suspension we obtained a volume of 0.15 L which means an 85% water removal that led to the formation of a gel-like suspension.”

Point 3: Page 10, line 340, “The Carreau-Yasuda rheological model” and line 341, “Herschel-Bulkley fitting model”: The authors should provide the definitions and sources of these models and discuss their application to the present experimental results in light of previous studies.

Response 3: We added the definitions and sources in the Materials and methods, subsection 4.5.6, lines 866-873). Additional application aspects and comparative discussions were provided in the Results and Discussion section (lines 483-488).

Point 4: Figures 7 and 8: In Figures 7b, 7c, 8b, and 8c, the values are noted as thixotropy, but what do they mean? The authors should provide a definition and source for thixotropy in a single data and discuss these results.

Response 4: We added the definitions and sources in the Materials and methods, subsection 4.5.6, lines 873-876). The thixotropic behavior was discussed first at line 451 and from line 466 to line 476.

Reviewer 3 Report

Comments and Suggestions for Authors

The manuscript is titled ‘Bacterial cellulose from Kombucha pellicle for mucoadhesive nanoformulations – a comparison with vegetal cellulose from brewer’s spent grains.’ This study aims to assess the efficacy of bacterial and plant-derived nanocellulose in the formation of hydrogels for biomedical applications. By the authors cellulose nanofibers derived from vegetal and bacterial sources were synthesized from brewer's waste grains and Kombucha membranes, respectively. However, it is essential to offer further clarification on specific issues. Consequently, it is advised that this paper be published after minor revisions.

In TEM images, what is the possible reason behind the edges of the fiber is darker? It was associated with thickness and amorphous structure, but it is not well explained why these dark areas are thicker or amorphous compared to other regions? Please explain it with related references.

Line 205 à The preparation methods and explanations of hydrogels unnecessary in Figure 5 caption.

Line 261 à Table 1 caption: The abbreviations or the explanations should used not both of them.

Line 288 à It is not “Figure 6a and b” it should be “Figure 7a and b”.

Author Response

The manuscript is titled ‘Bacterial cellulose from Kombucha pellicle for mucoadhesive nanoformulations – a comparison with vegetal cellulose from brewer’s spent grains.’ This study aims to assess the efficacy of bacterial and plant-derived nanocellulose in the formation of hydrogels for biomedical applications. By the authors cellulose nanofibers derived from vegetal and bacterial sources were synthesized from brewer's waste grains and Kombucha membranes, respectively. However, it is essential to offer further clarification on specific issues. Consequently, it is advised that this paper be published after minor revisions.

Point 1. In TEM images, what is the possible reason behind the edges of the fiber is darker? It was associated with thickness and amorphous structure, but it is not well explained why these dark areas are thicker or amorphous compared to other regions? Please explain it with related references.

Response 1: Thank you for the question. We don't know exactly to which edges in TEM images you are referring to. If you are referring to the edges from Figure 2b (top-right corner) and Figure 2c (bottom-left corner, bottom-right corner, top-right corner), those areas represent the lacey formvar/carbon type-B film applied on the 200 mesh cooper grid that we used for the TEM analysis. If you are referring to the darker areas of the cellulose nanofibers, these are optical effects induced by the thickness of the sample (if a region of the sample is thinner, it appears brighter, and if a region of the sample is thicker, it appears darker). We did comment on darker/lighter regions, but for the SEM images. We decided to remove this part as we do not have enough evidence to claim this aspect.

Point 2: Line 205 à The preparation methods and explanations of hydrogels unnecessary in Figure 5 caption. Line 261 à Table 1 caption: The abbreviations or the explanations should used not both of them.

Response 2: Thank you for your logical suggestion. However, the MDPI publisher in its layout style guide (https://www.mdpi.com/authors/layout#_bookmark14) mentions in Section 3.5 Abbreviations the following: Note that the abstract, main text and figure/table/diagram captions are treated separately for abbreviations. […] The reason for this is that they are often displayed in isolation; for example, indexing services usually display only the abstract and you can browse the figures without the main text via the journal's website.” Although MDPI does not request specifically abbreviation definition in each Figure and Table, it is considered as a general rule that each table and figure must be a stand-alone body and understood independently of other references, for several reasons (easy understanding of the figure, other authors could use the graphical display in future works and need to have the definitions of the terms attached etc.). Therefore, we consider that the figure / table caption is an important part for the understanding and the interpretation of the scientific data by all the readers and that it should contain every time the explanation for each abbreviation in the figure / table.

Point 3: Line 288 à It is not “Figure 6a and b” it should be “Figure 7a and b”.

Response 3: Thank you very much for your observation. We modified.

Reviewer 4 Report

Comments and Suggestions for Authors

1. The title needs to be modified and shortened

2. Please add more quantitative results to the abstract instead of a general explanation. those wow results.

3. Methods:

all methods were explained in detail. appropriate references were cited. all amounts and operational parameters were mentioned, Excellent.

4. Results and discussion:

I have a problem with the terms "nanocellulose" and "nano fibrillar " in line 119. when we claim nanofibrillar we must have separated fibers. in your images, you just zoomed in nano scale. there is no nano fibril. I also checked Ref|#29. they had microfibril not nanofibril. if you have microfibril then at first you should provide images of microfibril then correct the whole MS. Ref#41 is a good reference.

L141: I do not agree. does not make sense scientifically. 

L171: yes, it has a compact structure but is not separated. after a post-processing like acid-based approaches then you can produce micro or nanocellulose.

L233, L251. remove the dot.

L366: I wish you could predict a chemical structure for both BNC and VNC showing the main groups and also the arrangment. Are they different from this viewpoint?

L405-406: Why?

In the case of hydrogels, the rate of water uptake, porosity, swelling, and water retention are required tests. Why you did not measure them.

Author Response

Point 1: The title needs to be modified and shortened

Response 1: Thank you for your suggestion. We modified and shortened the title.

Point 2: Please add more quantitative results to the abstract instead of a general explanation. those wow results.

Response 2: Thank you for your suggestion. We included more quantitative information.

Point 3: Methods:

all methods were explained in detail. appropriate references were cited. all amounts and operational parameters were mentioned, Excellent.

Response 3: Thank you very much for your appreciation.

Point 4: Results and discussion:

I have a problem with the terms "nanocellulose" and "nano fibrillar " in line 119. when we claim nanofibrillar we must have separated fibers. in your images, you just zoomed in nano scale. there is no nano fibril. I also checked Ref|#29. they had microfibril not nanofibril. if you have microfibril then at first you should provide images of microfibril then correct the whole MS. Ref#41 is a good reference.

Response 4: Thank you for the comment. We partially agree with your comment. In the case of VNC the TEM micrographs at lower magnification indicates that VNC was apparently less fibrillated than BNC. BNC was however mostly fibrillated into nanofibers, as defined by the technological terminology. The diameter of most fibers is below 100 nm. We included a clarifying phrase at the beginning of the TEM results, respectively “Nanomaterials are defined as materials with at least one of their dimensions in the nanometric scale, i.e., ≤ 100 nm, the nano-dimension being their diameter [27-30], in our case 20–100 nm” (lines 148-149). We also included in Figure 2 the TEM images with the 2 µm scale. Nevertheless, it is also possible that some fibers aggregated upon hydrogel formation as previously shown. We added all these in the text and that we continued to use the term nanofibers for VNC as well (lines 148-159)

Point 5: L141: I do not agree. does not make sense scientifically. 

Response 5: Thank you for your observation. We removed this part as there are not enough proves to sustain it.

Point 6: L171: yes, it has a compact structure but is not separated. after a post-processing like acid-based approaches then you can produce micro or nanocellulose.

Response 6: Thank you for your comment. The paragraph starting from L171/172 is about hydrogel-mucin interaction. We removed the previous phrase related to similar fibrillation of VNC and BNC, as TEM micrographs at 2 µm shows a better fibrillation of BNC.

Point 7: L233, L251. remove the dot.

Response 7: Thank you for your observation. We removed them.

Point 8: L366: I wish you could predict a chemical structure for both BNC and VNC showing the main groups and also the arrangement. Are they different from this viewpoint?

Response 8: We determined by XRD the main allomorphs of the two types of cellulose, showing that the bacterial cellulose has a predominant peak for Iα allomorph and BSG cellulose for Iβ allomorph and amorphous cellulose. The main groups of cellulose and the crystal arrangements of these allomorphs have already been reported. We added this information and relevant citations (lines 235-237). We additionally performed FTIR analysis and found that the two types of celluloses have similar spectra, with some peak shifts and some differences in intensities inherent to differences in chain organization (lines 279-316). VNC seems to present more inter-molecular hydrogen interactions between the chains. The available data and techniques that we have at this moment do not allow a more specific and detailed prediction of the (chemical and/or crystal) structures.

Point 9: L405-406: Why?

Response 9: We tried to explain this decrease in the cell viability by the length and organization of the cellulose nanofibers, according to a reported study that investigated this particular issue. We also included at the beginning of the paragraph “By morphological analyses we highlighted that the BNC hydrogel exhibited a mesh-like structure, more fibrillated into relatively long fibers than VNC.” (lines 630-631). We determined additionally the porosity and found that VNC could not be analyzed probably due to a very low porosity which could also contribute to the effects observed. The low porosity could be related to a lower degree of fibrillation as suggested by TEM (lines 543-569). We hope we managed to clarify this aspect.

Point 10: In the case of hydrogels, the rate of water uptake, porosity, swelling, and water retention are required tests. Why you did not measure them.

Response 10: Thank you for your observation. Based on your suggestion to include the porosity of the two hydrogels, we performed and added the BET analysis (lines 543-569). With respect to the water uptake, swelling and water retention, we could not perform these analyses because our hydrogel disperses into water relatively fast, as it is not covalently/chemical cross-linked / reticulated.  It is a physical network hydrogel. Moreover, it is not meant to be applied in a lyophilized form (aerogel), but in the hydrated form.

Round 2

Reviewer 1 Report

Comments and Suggestions for Authors

The authors have satisfactorily addressed the reviewer's questions. I still think that the subchapters in chapter 2 could have been more divided but this is a matter of aesthetics. In this form the work can be published.